# ALIGNCHAT: ENDOWING LLMS WITH END-TO-END SPEECH-TO-TEXT CHAT CAPABILITY THROUGH TOKEN-LEVEL REPRESENTATION ALIGNMENT

## ABSTRACT

The advent of large multimodal models (LMMs) such as GPT-4o has intensified interest in equipping large language models (LLMs) with end-to-end speech understanding capabilities. Existing methods typically employ encoder-based audio tokenizers to map speech into audio tokens served as LLM inputs; while effective, the frequency discrepancy between audio and text tokens demands large quantities of speech data and costly LLM finetuning to achieve cross-modality alignment, while potentially harming the original capability of the LLM backbone. In this work, we introduce *AlignChat*, a simple yet effective framework that bridges speech and text modality via a speech tokenizer with encoder-decoder-based Transformer architecture, ensuring precise one-to-one token-level alignment and efficient cross-modality knowledge transfer without the need for finetuning the LLM backbone. AlignChat adopts a two-stage training scheme. The computation-efficient pretraining stage only requires the speech tokenizer and embeddings of the LLM for preliminary cross-modality alignment, while the instruction-tuning stage proposes to use self-generated speech-instruction-response pairs to ensure consistency between speech- and text-conditioned behavior of AlignChat. Experiments demonstrate that AlignChat achieves strong performance on speech-to–text chat benchmarks with only ~1/20 of the speech data used by previous methods.

## 1 INTRODUCTION

Speech is a fundamental form of human communication and is critical in real-world applications. With the rapid advancement of large language models (LLMs) (OpenAI, 2023; Dubey et al., 2024; DeepSeek-AI et al., 2025), developing voice assistants capable of efficient speech understanding has become a heated research goal. Early cascaded pipelines (Huang et al., 2024c) transcribe speech via automatic speech recognition (ASR) before processing it with an LLM, but suffer from high latency and loss of salient speech characteristics such as timbre and emotion. More recently, proprietary large multimodal models (LMMs) such as GPT-4o (Hurst et al., 2024) have demonstrated seamless, low-latency, end-to-end speech interaction capability, motivating open-source counterparts.

Recent open-source efforts (Xu et al., 2025; Li et al., 2025b; Chen et al., 2025) have sought to leverage LLMs for speech understanding by introducing pretrained audio encoders with extra modality projectors to transform speech into temporally uniform audio tokens compatible with text-trained LLMs. However, this approach faces a fundamental challenge: audio and text tokens *differ markedly* in frequency characteristics (e.g., a single word corresponds to ~1 text token but ~5 audio tokens (Chen et al., 2025)), making alignment within shared representation space inherently difficult. Bridging this gap typically requires training on massive quantities of speech data (>100 K hours) (KimiTeam et al., 2025), and often necessitates finetuning the LLM (Huang et al., 2024b). Unfortunately, in practice, finetuning on speech–dialogue corpora comprising short, syntactically simple sentences can degrade the LLM's original performance (see Table 5, Appendix) (Chen et al., 2024b), a phenomenon also known as *catastrophic forgetting* (Kirkpatrick et al., 2017).

To address these challenges, we propose *AlignChat*, an efficient and robust framework that enforces precise one-to-one token-level alignment between speech and text modality by explicitly modeling their correspondence. This design enables speech tokens to fully align with LLM's embedding space

while preserving non-textual features such as prosody and speaker characteristics, thereby avoiding the information loss inherent to the non-end-to-end ASR–LLM cascaded pipelines.

**Model architecture.** AlignChat follows the general design of other LMMs by comprising a speech tokenizer, a projector, and an LLM. To achieve one-to-one alignment between modalities, we adopt an encoder–decoder speech tokenizer. Speech inputs are encoded into audio features at high frequency with the encoder, while the decoder autoregressively reduces speech token frequency to establish correspondence with text tokens. Crucially, the speech tokenizer shares LLM's vocabulary, producing speech representations directly aligned with LLM's input space for seamless integration.

**Training strategy.** Training proceeds in two stages. In the first stage, the decoder and projector are pretrained on speech recognition datasets to align speech representations with the LLM's input embedding space. In the second stage, we use speech–dialogue datasets for end-to-end instruction tuning, yet we update only the decoder and projector while keeping the LLM frozen. This approach maintains LLM's original language abilities while efficiently incorporating speech understanding.

**Data synthesis.** To ensure that introducing speech tokens does not disrupt the LLM's original behavior and complex abilities, we utilize publicly available speech datasets to regenerate new speech-instruction-response pairs using the LLM itself. This guarantees consistency between speech- and text-conditioned behavior of AlignChat and further mitigates risks of performance degradation.

To the best of our knowledge, AlignChat is the first approach to employ an *encoder–decoder architecture for strict token-level alignment* to integrate speech understanding capabilities into LLM while keeping LLM parameters entirely frozen. This design preserves complex abilities such as reasoning and instruction following, resulting in an efficient and robust cross-modality framework.

Extensive experiments show that AlignChat achieves strong performance with only ~1/20 of the data required by prior works. Notably, AlignChat outperforms prior fully end-to-end state-of-the-art, Kimi-Audio, by 0.38% on OpenAudioBench and by 1.8% on VoiceBench. Large gains are observed in speech understanding (+7.48% on MMSU), helpfulness (+5.53% on AlpacaEval), and instruction-following (+2.07% on IFEval). Further experiments confirm that AlignChat preserves key non-textual features after alignment training, distinguishing from non-end-to-end cascaded pipelines.

## 2 ALIGNCHAT

As illustrated in Figure 1, AlignChat adopts a fully end-to-end paradigm for speech-to–text language modeling. Speech inputs are transformed into continuous representations strictly aligned with the LLM's input text embeddings via an encoder–decoder–based Transformer architecture and a modality projector. The aligned representations are then fed into the frozen LLM backbone to produce end-to-end text responses. Only the modality projector and decoder part of the speech tokenizer are trainable, while all other modules remain frozen to preserve the capability in the pretrained models. This section details the encoder–decoder speech tokenizer (Section 2.1), the token-level representation alignment strategy (Section 2.2), the multi-stage training procedure (Section 2.3), and the instruction-tuning alignment data synthesis (Section 2.4).

### 2.1 ENCODER-DECODER SPEECH TOKENIZER

AlignChat employs an encoder–decoder speech tokenizer initialized with Whisper (Radford et al., 2023) to convert speech into compact, token-level aligned representations. Given speech of $t$ seconds, AlignChat first uses the speech encoder to produce intermediate audio features $\boldsymbol{A}$ at 50 Hz:

$$\boldsymbol{A} = \text{Encoder}(\textit{Speech Inputs}) = [\boldsymbol{a}_1, \boldsymbol{a}_2, \ldots, \boldsymbol{a}_{L_{\text{audio}}}] \in \mathbb{R}^{L_{\text{audio}} \times d_{\text{audio}}}, \tag{1}$$

where $L_{\text{audio}} = 50 \cdot t$ and each $\boldsymbol{a}_i \in \mathbb{R}^{d_{\text{audio}}}$ is an extracted audio feature. This 50 Hz frequency of audio features greatly exceeds that of text embeddings (~3 Hz), leading to an ~16× length mismatch between speech and text token sequences with same content. This disparity increases computational cost and hinders cross-modality capability transfer. Prior works (KimiTeam et al., 2025; Xu et al., 2025) partially alleviate this by downsampling audio features to ~10 Hz, but the mismatch persists.

Instead of downsampling and feeding audio features $\boldsymbol{A}$ into the LLM like prior methods, AlignChat passes audio features first through a speech decoder to obtain more compact representation sequence, $\boldsymbol{S} = [\boldsymbol{s}_1, \boldsymbol{s}_2, \ldots, \boldsymbol{s}_{L_{\text{speech}}}] \in \mathbb{R}^{L_{\text{speech}} \times d_{\text{audio}}}$, that is strictly aligned with LLM's input embeddings.

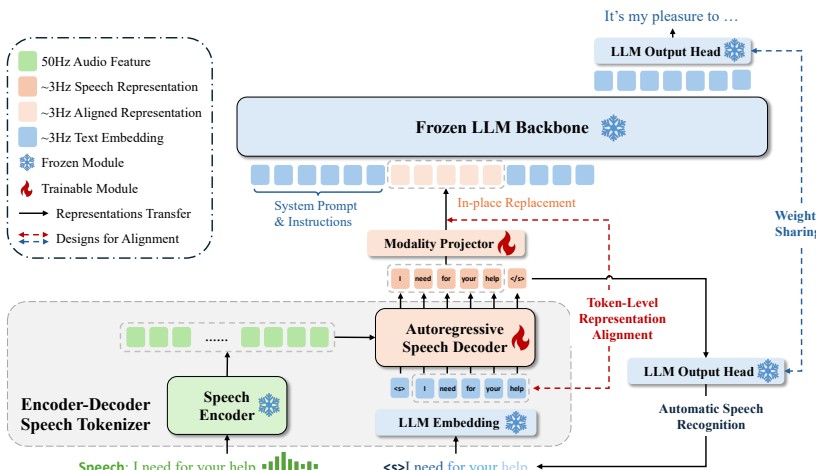

Figure 1: Architecture of AlignChat. Speech inputs are first processed by a speech encoder to extract audio features. These features are then autoregressively decoded by a Transformer-based speech decoder, producing both ASR outputs and speech representations. Each speech representation is projected and then strictly aligned one-to-one with a text embedding, enabling direct substitution with the LLM's input embeddings to generate the final end-to-end response. Throughout training, the speech encoder and LLM backbone remain frozen to preserve their pretrained audio feature extraction and language modeling capabilities. The embedding layer and output head are shared between the speech decoder and the LLM backbone for efficient capability transfer between modalities.

Since Whisper and the LLM backbone have different vocabularies, the outputs of original Whisper decoder do not exhibit strict one-to-one correspondence with the LLM's input embeddings. They thus cannot be directly used by the LLM backbone. As shown in Figure 1, AlignChat proposes to share the vocabulary and embedding layer between the speech decoder and LLM backbone, aligning tokenization across modalities. Let the text corresponding to the speech inputs be tokenized into $\boldsymbol{T} = [\text{token}_1, \ldots, \text{token}_{L_{\text{text}}}]$ with embeddings $\boldsymbol{X}^{\text{text}} = [\boldsymbol{x}_1^{\text{text}}, \ldots, \boldsymbol{x}_{L_{\text{text}}}^{\text{text}}] \in \mathbb{R}^{L_{\text{text}} \times d_{\text{llm}}}$. AlignChat ensures that $L_{\text{speech}} = L_{\text{text}} \ll L_{\text{audio}}$ and representation pair $(\boldsymbol{s}_i, \boldsymbol{x}_i^{\text{text}})$ corresponds to the same $\text{token}_i$. We can thus define the compact sequence of speech representations output by the autoregressive speech decoder (dimension-matched to the LLM via learned projections[1]) as:

$$\boldsymbol{S} = \text{Decoder}(\boldsymbol{A}, \boldsymbol{X}^{\text{text}}) = [\boldsymbol{s}_1, \ldots, \boldsymbol{s}_{L_{\text{text}}}] \in \mathbb{R}^{L_{\text{text}} \times d_{\text{llm}}}. \tag{2}$$

AlignChat uses the derived speech representations $\boldsymbol{S}$ to jointly perform speech transcription and produce aligned representations that can be replaced in-place to the LLM's input embeddings for end-to-end text response (Figure 4, Appendix). The speech decoder and modality projector are fine-tuned after vocabulary/embedding sharing, which will be described in Section 2.2 and Section 2.3.

## 2.2 TOKEN-LEVEL REPRESENTATION ALIGNMENT

Speech is unique among modalities (e.g., image, video) in that it can be transcribed into text with a strict one-to-one correspondence. AlignChat exploits this to establish token-level representation alignment. As shown in Figure 1, a modality projetor $\text{MLP}_{\text{align}}(\cdot)$, implemented as a two-layer perceptron, transforms speech representations $\boldsymbol{S}$ into new representations $\boldsymbol{X}^{\text{align}}$:

$$\boldsymbol{X}^{\text{align}} = [\boldsymbol{x}_1^{\text{align}}, \ldots, \boldsymbol{x}_{L_{\text{text}}}^{\text{align}}] = \text{MLP}_{\text{align}}(\boldsymbol{S}), \tag{3}$$

aligned to LLM's input space and can thus serve as inputs to the frozen LLM backbone. The modality projector decouples transcription from representation alignment to avoid cross-task interference.

A perfect alignment between speech and text representations would make $\boldsymbol{X}^{\text{align}} = \boldsymbol{X}^{\text{text}}$, achievable in principle by minimizing direct matching losses such as mean absolute error (MAE). However, non-textual features (e.g., emotion) should be preserved in speech representations, so achieving

---

[1]Dimensions of Whisper and the LLM backbone are different ($d_{\text{audio}} \neq d_{\text{llm}}$). We use linear projections $\boldsymbol{W}_{\text{proj\_in}} \in \mathbb{R}^{d_{\text{llm}} \times d_{\text{audio}}}$ and $\boldsymbol{W}_{\text{proj\_out}} \in \mathbb{R}^{d_{\text{audio}} \times d_{\text{llm}}}$ to match dimensions across pretrained models.

Table 1: List of all datasets used for AlignChat training. The *Stage* column shows the training stage the dataset is used in, where Stage 1 is pretraining and Stage 2 is instruction tuning.

| Dataset | #Hours | #Samples | Subset | Task | Use Stage |
|---|---|---|---|---|---|
| MLS (Pratap et al., 2020) | 10,000 | 2,420,047 | eng_10k | ASR | 1 |
| CommonVoice (Ardila et al., 2020) | 1,787 | 1,131,718 | en | ASR | 1 & 2 |
| LibriTTS-R (Koizumi et al., 2023) | 542 | 339,167 | train | ASR | 1 & 2 |
| Eurospeech (Pfisterer et al., 2025) | 169 | 40,791 | uk | ASR | 1 & 2 |
| Fleurs (Conneau et al., 2022) | 7.26 | 2,580 | en_us | ASR | 1 & 2 |
| VoiceAssistant-400K (Xie & Wu, 2024) | 683 | 470,054 | all | Dialogue | 1 & 2 |
| DeepDialogue Koudounas et al. (2025) | 534 | 243,279 | all | Dialogue | 1 & 2 |
| Spoken Web Questions (Ultravox, 2025) | 2.42 | 3,778 | train | QA | 2 |
| OpenBookQA[†](Mihaylov et al., 2018) | 22.39 | 4,957 | train | MCQ | 2 |

[†] We synthesize questions from OpenBookQA into speech for training with OpenVoice (Qin et al., 2023).

near-zero direct matching error is not fully satisfactory. Inspired by interpretability studies showing that embedding direction encodes semantics (Zou et al., 2023a), AlignChat augments direct matching with cosine similarity loss to achieve semantic alignment while allowing residual differences reserved for non-textual features. Since content-rich tokens (e.g., *Purgatorio*) appear less frequently than common tokens (e.g., *what*, *the*, *a*), we square the cosine similarity loss to penalize poorly aligned tokens. The resulting alignment loss is a combination of direct and semantic matching:

$$\mathcal{L}_{\text{align}} = \alpha \cdot \mathcal{L}_{L1} + \beta \cdot \mathcal{L}_{\text{cos\_sim}}^2 \tag{4}$$

$$= \alpha \cdot \text{L1}(\boldsymbol{X}^{\text{align}}, \boldsymbol{X}^{\text{text}}) + \beta \cdot \left(1 - \cos\_\text{sim}(\boldsymbol{X}^{\text{align}}, \boldsymbol{X}^{\text{text}})\right)^2, \tag{5}$$

where $\alpha$ and $\beta$ balance between alignment objectives; unless otherwise stated, AlignChat uses $\alpha = 1.0$ and $\beta = 5.0$ for both pretraining and instruction tuning stage emprically.

## 2.3 MULTI-STAGE TRAINING STRATEGY

AlignChat employs a two-stage training scheme, leveraging three types of loss (Figure 4, Appendix). Stage 1 (Pretraining) adapts speech decoder and modality projector to LLM's vocabulary and embeddings, while establishing initial representation alignment. Stage 2 (Instruction tuning) uses instruction–response pairs generated by the LLM itself to refine alignment in an end-to-end manner.

**Stage 1: transcription + alignment** An automatic speech recognition loss $\mathcal{L}_{\text{asr}}$, same as Whisper, restores the transcription capability of the speech tokenizer after vocabulary and embedding sharing described in Section 2.1. An alignment loss $\mathcal{L}_{\text{align}}$ (Section 2.2) constrains speech representations to align with corresponding text embeddings in the LLM's input space. The Stage-1 objective is:

$$\mathcal{L}_{\text{Stage\_1}} = \mathcal{L}_{\text{asr}} + \mathcal{L}_{\text{align}}. \tag{6}$$

The speech encoder is frozen to preserve audio feature extraction capability and accelerate convergence as Ma et al. (2024), while the speech decoder and modality projector $\text{MLP}_{\text{align}}(\cdot)$ are adapted to shared vocabulary and LLM's embeddings. Notably, *transformer layers from the LLM are not used in this pretraining stage*, significantly reducing computational resource requirements.

**Stage 2: end-to-end instruction tuning + transcription + alignment** An end-to-end language modeling loss $\mathcal{L}_{\text{llm}}$ is applied to the LLM's output logits given speech inputs. While $\mathcal{L}_{\text{align}}$ enforces numerical similarity between speech and text representations, $\mathcal{L}_{\text{llm}}$ supplies supervised, sequence-level, end-to-end gradients that prioritize aligning dimensions crucial for the LLM's text generation functionality. The $\mathcal{L}_{\text{asr}}$ is used to preserve speech transcription capability. The Stage 2 objective is:

$$\mathcal{L}_{\text{Stage\_2}} = \mathcal{L}_{\text{llm}} + \mathcal{L}_{\text{asr}} + \mathcal{L}_{\text{align}}. \tag{7}$$

The speech encoder and LLM backbone are still kept frozen in this stage, while the speech decoder and modality projector $\text{MLP}_{\text{align}}(\cdot)$ remain trainable. The instruction tuning data is designed to ensure consistent LLM behavior given speech and text inputs conveying identical semantic meanings. The alignment data synthesis process is described in Section 2.4.

While multi-stage training is commonly adopted by LMMs (Wang et al., 2024; Xie & Wu, 2024; Fu et al., 2025), AlignChat differs by requiring *only the embedding layer and output head from the*

*LLM backbone for alignment in the pretraining stage.* Aligning the speech tokenizer to the LLM representation space without a forward pass of the LLM reduces computational overhead compared with prior approaches. Experiments (Section 3.3.1) show that AlignChat already achieves strong performance after the pretraining stage, while the instruction tuning stage delivers further gains.

## 2.4 ALIGMENT DATA SYNTHESIS

Previous studies (Yan et al., 2024; Cho et al., 2024) have observed that speech responses in human dialogues tend to be shorter, simpler, and limited to content easily verbalized, compared to text-based responses. Existing datasets (Xie & Wu, 2024; Koudounas et al., 2025) are meticulously curated to reflect these characteristics, adapting LLMs for speech dialogues. However, directly finetuning LLMs on simplified instruction-response pairs may significantly undermine performance in complex tasks, such as reasoning and instruction following (Yang et al., 2024; Chen et al., 2024a).

AlignChat instead seeks to preserve the LLM backbone's *original text response competence* by aligning its behavior on speech and corresponding text inputs. Specifically, original responses in the speech dialogue dataset are discarded: for each speech input, the paired text is fed into the LLM backbone to generate the response representing its original behavior. These responses maintain the sophistication of text-based outputs while enabling consistent cross-modal alignment, producing self-generated instruction–response pairs used in instruction tuning. Although AlignChat does not explicitly adapt the LLM backbone to the speech-specific response rules outlined above, modern LLMs' role-playing and style adaptation capabilities are sufficient to satisfy these requirements.

As summarized in Table 1, AlignChat uses publicly available datasets in all training stages. Existing ASR datasets are more diverse and longer than speech QA datasets. Still, transcripts in ASR datasets are often incomplete text fragments that fail to elicit meaningful LLM responses[2]. To address this, AlignChat manually designs 40 instructions targeting specific textual elements, e.g., *"Extract the nouns from the following line and provide explanations"*, to generate training samples strengthening alignment using transcripts in ASR datasets. Full prompt templates are provided in Appendix A.5.3.

## 3 EXPERIMENTS

### 3.1 EXPERIMENTS SETUP

#### 3.1.1 TRAINING AND INFERENCE DETAILS

We use *Qwen2.5-7B-Instruct* as the frozen LLM backbone for fair comparison with state-of-the-art end-to-end models, such as Kimi-Audio (KimiTeam et al., 2025) and Qwen2.5-Omni (Xu et al., 2025), initialized with the same backbone. The Qwen chat template and special tokens are used without modification. Notably, AlignChat only requires representation alignment, which is agnostic to the LLM's internal architecture and can be adapted to models of different sizes and architectures.

The encoder–decoder speech tokenizer is initialized from *whisper-large-v3-turbo* and finetuned using the two-stage strategy described in Section 2.3. Instead of using the same audio tokenizer, *whisper-large-v3*, as previous works, AlignChat uses the turbo variant to accommodate the increased computation and parameter overhead brought by the additional decoder.

Both pretraining and instruction tuning run for two epochs. We use the AdamW optimizer Loshchilov & Hutter (2019) with a maximum learning rate of $1 \times 10^{-4}$, a linear warmup at first $5\%$ steps, and cosine annealing to a minimum of $1 \times 10^{-6}$. The global batch size is 256 for pretraining and 96 for instruction tuning, with gradient accumulation used when GPU memory is insufficient.

During inference, speech inputs are first encoded into audio features and then autoregressively converted into speech representations that maintain a strict correspondence with text tokens. These representations are subsequently mapped into LLM's input space via the modality projector and directly substituted into the user-content slot of the chat template. The resulting sequence is then passed to the LLM backbone to generate end-to-end text responses. For reproducibility, we adopt greedy decoding for both the generation of speech representations and the final text outputs.

---

[2]For example, given an excerpt from *Purgatorio: Canto Twenty-nine*, the LLM may output *"Sorry, I don't understand ... Could you please provide more information?"*, which does not support effective alignment.

Table 2: Performance of AlignChat and baseline models on the speech-to-text chat tasks. Results of different subsets are separated by vertical lines |. Each subset's best and second-best results are in **bold** and underline respectively. AlignChat achieves strong performance on OpenAudioBench and VoiceBench using an order of magnitude less *speech* training data than previous works.

| Datasets | Model | #kHours | Performance (↑) | | | | |
|---|---|---|---|---|---|---|---|
| **OpenAudioBench** AlpacaEval \| Llama Q. \| TriviaQA \| Web Q. \| Avg. | Qwen2-Audio | 370 | 57.19 | 69.67 | 40.30 | 45.20 | 53.09 |
| | Baichuan-Audio | 185 | 59.65 | 74.33 | 55.40 | 58.70 | 62.02 |
| | GLM-4-Voice | 710 | 57.89 | 76.00 | 51.80 | 55.40 | 60.27 |
| | Freeze-Omni[†] | 110 | – | 72.00 | 53.88 | 44.73 | – |
| | Step-Audio | 650 | 56.53 | 72.33 | 56.80 | **73.00** | 64.67 |
| | Baichuan-Omni-1.5 | 206 | 77.90 | 78.50 | 57.20 | 59.10 | 68.12 |
| | Qwen2.5-Omni | – | 72.76 | 75.33 | 57.06 | 62.80 | 66.99 |
| | Kimi-Audio | 290 | 75.73 | 79.33 | **62.10** | 70.20 | 71.84 |
| | *AlignChat* | 15 | **81.26** | **80.33** | 61.40 | 65.90 | **72.22** |
| **VoiceBench** AlpacaEval \| CommonEval \| WildVoice \| SD-QA \| MMSU | Qwen2-Audio | 370 | 3.74 | 3.43 | 3.01 | 35.71 | 35.72 |
| | Baichuan-Audio | 185 | 4.41 | 4.08 | 3.92 | 45.84 | 53.19 |
| | GLM-4-Voice | 710 | 3.97 | 3.42 | 3.18 | 36.98 | 39.75 |
| | Freeze-Omni | 110 | 4.03 | 3.46 | 3.15 | 53.45 | 28.14 |
| | Step-Audio | 650 | 4.13 | 3.09 | 2.93 | 44.21 | 28.33 |
| | Baichuan-Omni-1.5 | 206 | 4.50 | 4.05 | 4.06 | 43.40 | 57.25 |
| | Qwen2.5-Omni | – | 4.49 | 3.93 | 3.83 | 55.71 | 61.32 |
| | Kimi-Audio | 290 | 4.46 | 3.97 | 4.20 | **63.12** | 62.17 |
| | *AlignChat* | 15 | **4.57** | **4.13** | **4.21** | 62.84 | **69.65** |
| **VoiceBench** OpenBookQA \| BBH \| IFEval \| AdvBench \| Avg. | Qwen2-Audio | 370 | 49.45 | 54.70 | 26.33 | 96.73 | 55.80 |
| | Baichuan-Audio | 185 | 71.65 | 54.80 | 50.31 | 99.42 | 69.27 |
| | GLM-4-Voice | 710 | 53.41 | 52.80 | 25.92 | 88.08 | 56.48 |
| | Freeze-Omni | 110 | 30.98 | 50.70 | 23.40 | 97.30 | 55.20 |
| | Step-Audio | 650 | 33.85 | 50.60 | 27.96 | 69.62 | 50.84 |
| | Qwen2.5-Omni | – | 81.10 | 68.50 | 52.87 | 99.42 | 73.77 |
| | Kimi-Audio | 290 | 83.52 | **69.70** | 61.10 | **100.0** | 76.91 |
| | *AlignChat* | 15 | **85.49** | 69.60 | **63.17** | 99.42 | **78.71** |

[†] Different from others, Freeze-Omni uses *edge-tts* to synthesize text questions in subsets into speech.

### 3.1.2 BASELINES, BENCHMARKS AND METRICS

We compare AlignChat against open-source models with speech-to-text chat capability, including Qwen2-Audio (Chu et al., 2024), Baichuan-Audio (Li et al., 2025a), GLM-4-Voice (Zeng et al., 2024), Freeze-Omni (Wang et al., 2024), Step-Audio (Huang et al., 2025), Baichuan-Omni-1.5 (Li et al., 2025b), Qwen2.5-Omni (Xu et al., 2025), and Kimi-Audio (KimiTeam et al., 2025).

To assess the speech-to-text chat performance of AlignChat, we use open-source benchmarks widely adopted by previous works: OpenAudioBench (Li et al., 2025b) and VoiceBench (Chen et al., 2024b). Each of these benchmarks contains several subsets that evaluate different dimensions of capabilities: helpfulness (AlpacaEval (Li et al., 2023)), knowledge (Llama Questions (Nachmani et al., 2024), etc.), instruction following (IFEval (Zhou et al., 2023)), and safety (AdvBench (Zou et al., 2023b)). Together, these form a comprehensive evaluation of speech-to–text chat capability.

We adopt the official evaluation protocols in the corresponding benchmarks. For open-ended QA, advanced LLMs (e.g., GPT-4o (OpenAI, 2023)) are used to rate the model responses. For reference-based QA, responses and ground-truth answers are jointly provided to advanced LLMs to evaluate correctness. For multiple-choice QA, the predicted option is compared against the ground truth. Instruction-following is measured by adherence to specific requirements, and safety by the rejection rate of harmful instructions. Prompts used for evaluation are in Appendix A.5.2.

### 3.2 MAIN RESULTS

Speech-to-text chat capability evaluation results of AlignChat and baselines are shown in Table 2.

On **OpenAudioBench**, AlignChat achieves superior results on several subsets, including AlpacaE-val and Llama Questions, while remaining competitive on TriviaQA and Web Questions. The largest gain appears on AlpacaEval (81.26 *vs.* 75.73), which measures helpfulness via open-ended QA and tests complex abilities beyond reference-based QA. Alongside Table 5 (Appendix), this supports our observation that training solely on simple speech-dialogue data degrades LLM's pure-text abilities.

On **VoiceBench**, the trend is similar: AlignChat outperforms baselines on AlpacaEval (different questions than in OpenAudioBench), CommonEval, WildVoice, MMSU, OpenBookQA, and IFE-val. AlignChat shows strong performance on IFEval, which probes instruction-following capability, where prior works Yang et al. (2024); Chen et al. (2024a) report severe degradation after multimodal training. In contrast, AlignChat fully retains the LLM backbone's instruction-following skill (since it's frozen). Performance is competitive on SD-QA, BBH, and AdvBench, with slightly lower results likely due to low-frequency words underrepresented in the training datasets shown in Table 1.

Notably, AlignChat achieves these results with *an order of magnitude less training data* than many baselines. For example, Kimi-Audio uses ~235 K hours of open-source and ~55 K hours of in-house ASR data, whereas AlignChat trains on just ~15 K hours of open-source ASR, speech dialogue, and QA data. This efficiency stems from freezing the LLM backbone to avoid catastrophic forgetting and leveraging strict token-level alignment to transfer text capabilities to speech effectively.

### 3.3 ABLATION AND DISCUSSION

Table 3: Ablation experiments of AlignChat on OpenAudioBench. Higher value means better performance. Best and second-best results are in **bold** and underline.

| Setting | Alpaca. | LlamaQ. | Trivia. | Web Q. | Avg. |
|---|---|---|---|---|---|
| *AlignChat* | **81.26** | 80.33 | 61.40 | **65.90** | **72.22** |
| w\o pretrain | 79.75 | 79.67 | 58.10 | 61.80 | 69.83 |
| w\o fine-tune | 80.65 | **81.00** | 60.00 | 60.90 | 70.64 |
| w\o alignment | 79.15 | 78.67 | 61.30 | 63.90 | 70.67 |
| w\o decoder | 80.55 | 80.00 | **61.50** | 65.70 | 71.94 |

Table 4: Probing accuracy of gender and emotion on the speech representations derived from AlignChat.

| Split | Gender (Acc.) male *vs.* female | | Emotion (Acc.) pos. *vs.* neg. | |
|---|---|---|---|---|
| | 1st tok. | last tok. | 1st tok. | last tok. |
| train | 98.88 | 100.00 | 90.47 | 94.38 |
| test | 91.00 | 95.75 | 79.50 | 86.25 |

#### 3.3.1 TOKEN-LEVEL REPRESENTATION ALIGNMENT ENHANCES PERFORMANCE

We ablate major components of AlignChat to assess the individual contributions as shown in Table 3.

We analyze the *AlignChat* row in comparison with the ablated variants. Without pretraining or fine-tuning, the model retains strong performance on AlpacaEval and Llama Questions, which contain relatively common words and descriptive content, but degrades on TriviaQA and Web Questions, which involve more diverse and knowledge-intensive queries. However, the obtained models remain competitive with prior work. This suggests that token-level representation alignment alone provides effective transfer from text to speech, while end-to-end finetuning further boosts performance. Excluding the alignment loss consistently reduces performance, showing the necessity of token-level alignment as an auxiliary training signal.

The variant without speech decoder, which relies solely on the Whisper encoder for speech pre-processing, achieves slightly inferior performance than AlignChat, while introducing some practical limitations: (1) the downsampled Whisper encoder outputs still have a higher token frequency (12.5 Hz *vs.* ~3 Hz) than AlignChat, increasing LLM backbone inference cost especially for longer speech; and (2) the absence of intermediate text transcription and alignment harms interpretability, safety, and full multimodal experience (Huang et al., 2024b; Zhang et al., 2025).

AlignChat doesn't modify the internal structure of the LLM backbone, thus can be easily adapted to LLM backbones with different sizes. Please refer to Appendix A.2.3 for further results and analysis.

#### 3.3.2 NON-TEXTUAL FEATURES ARE PRESERVED IN ALIGNED REPRESENTATIONS

At the end of the AlignChat training, the token-level representations alignment loss is *close to but not strictly zero*. We use probing experiments to show that one of the reasons behind the discrepancy

of speech and text representations is that specific non-textual features (e.g., speaker's gender and emotion) are still preserved in the speech representations derived from AlignChat.

We sample 4000 speech inputs from the DeepDialogue (Koudounas et al., 2025) dataset, which contains meta information of the speaker's gender and emotion. The speech inputs are transformed to speech representations through the encoder-decoder speech tokenizer, which are then divided into 3200 training samples and 800 test samples. We train linear classifiers to perform gender (male *vs.* female) and emotion polarity (positive *vs.* negative) classification on the speech representations.

As shown in Table 4, gender and emotion classification achieve high accuracy (95.75% for gender and 86.25% for emotion) compared to random guessing (~50%), meaning that non-textual features corresponding to the speaker's gender and emotion are preserved after alignment training. We use the classification accuracy on the first speech token to indicate the preservation of non-textual features without the interference of text semantics. The accuracy is higher on the last token than on the first token on both gender and emotion classification tasks, showing that the non-textual features and text semantics are both gradually combined into speech representations through the speech decoder.

Previous interpretability works Zou et al. (2023a) have found that it's possible to change the response of LLMs through altering their internal hidden states. We discovered similar results that the responses of LLMs can be controlled by simply mixing the input text token embeddings with emotion token embeddings, which shows the potential of training empathetic end-to-end speech models without tuning LLM parameters. Examples of the comparisons of LLM responses when inputs are mixed with positive or negative emotions can be found in Appendix A.2.2.

### 3.3.3 ALIGNCHAT ACHIEVES WELL-ALIGNED CROSS-MODALITY REPRESENTATIONS

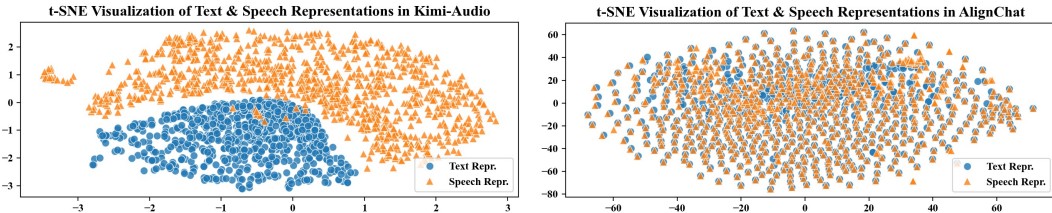

Figure 2: Visualization of speech and text representations from Kimi-Audio and AlignChat.

We extract speech and text representations corresponding to the same meaning from questions in OpenAudioBench using Kimi-Audio and AlignChat, and visualize them with t-SNE (Figure 2).

For Kimi-Audio, the visualization shows a pronounced separation between speech and text representation clusters. We ascribe this to the lack of an explicit alignment objective and the significant frequency mismatch between speech tokens and text tokens (12.5 Hz *vs.* ~3 Hz). This separation likely hinders the effective transfer of text capabilities to other modalities (Huang et al., 2024b).

By contrast, AlignChat achieves tight clustering of speech and text representations with same semantic content in the t-SNE space, reflecting effective cross-modality alignment. Meanwhile, AlignChat still shows distinguishable variation between speech and text representations, indicating the aligned speech representations preserve certain non-textual features potentially useful for better speech understanding (see examples in Appendix A.2.6) and building empathetic speech language models.

### 3.3.4 LIMITATIONS

**End-to-end empathetic training**    Previous works (Wang et al., 2025a; Huang et al., 2024a; Wu et al., 2025) have explored the empathetic ability in both language and speech models. An empathetic model should provide suitable responses according to user's emotional state and situation in both text and speech (e.g., emotion, style, timbre, tone). While AlignChat focuses on the behavior alignment of the LLMs when facing inputs of different modalities, we show through experiments (Section 3.3.2) that non-textual features are still preserved in the aligned speech representations, demonstrating the potential of incorporating further empathetic abilities. Efforts should be made to carefully synthesize data that can elicit empathy ability of LLMs without harming general abilities.

**Support general audio inputs** The core intuition of AlignChat is the strict correspondence between speech and text, which no longer holds for general audio inputs. A possible way is to use the existing audio captioning datasets to teach the model to align the representation of general audio to the corresponding text descriptions, e.g., *A woman talks nearby as water pours* and *The wind is blowing, insects are singing, and rustling occurs*. Alternatively, pretrained semantic tokens (Hsu et al., 2021) or acoustic tokens (Siuzdak et al., 2024) can be added to the LLM tokenizer and embedding layer. We plan to explore supporting general audio inputs in the following work.

# 4 RELATED WORKS

**Modality-Alignment Methods.** With the advent of powerful LLMs, substantial research has focused on extending them to handle additional modalities through alignment techniques (Zhu et al., 2024; Zhang et al., 2023; Hu et al., 2025). A common approach employs pretrained modality encoders (e.g., CLIP (Radford et al., 2021) and Whisper (Radford et al., 2023)) and connects them to LLM backbones via lightweight projectors (Zhu et al., 2024; Li et al., 2022). For example, SLAM-ASR(Ma et al., 2024) adopted the idea to perform ASR and demonstrated competitive performance by training only the lightweight projector. Instruction-following data, either generated by large proprietary models such as GPT-4o (Hurst et al., 2024) or open-source LLMs (Dubey et al., 2024), has been shown to improve end-to-end multimodal interaction fidelity (Liu et al., 2023; Lu et al., 2025). AlignChat differs from previous works by using an encoder-decoder speech tokenizer with cross-attention to align speech and text explicitly with a token-level representation alignment loss, and leverages self-generated instruction-tuning data for better end-to-end alignment.

**Speech-oriented Voice Assistants.** Several works (Xie & Wu, 2024; Défossez et al., 2024; KimiTeam et al., 2025; Chu et al., 2024; Li et al., 2025a; Kong et al., 2024) have developed voice assistants capable of real-time, speech-based interaction. Mainstream approaches preprocess speech with a Whisper encoder (Radford et al., 2023) and then map obtained features into the LLM's representation space, enabling end-to-end speech interactions. The LLM parameters are typically trainable for better performance. Other studies (Wang et al., 2024; Ultravox, 2024; Lu et al., 2025) explore supporting speech understanding while freezing the LLM backbone to avoid *catastrophic forgetting*. Benchmarks (Li et al., 2025b; Chen et al., 2024b; Wang et al., 2025b) targeting the evaluation of the capability of speech-oriented voice assistants are also emerging. Unlike prior work, AlignChat enforces a one-to-one mapping between speech and text tokens in LLM's input space, which naturally enables freezing the LLM's parameters to retain its text competence fully.

**Omni-modal Large Models.** Recent proprietary systems such as GPT-4o have inspired research into open-source omni-modal models that integrate understanding and generating text, speech, image, and video within a single unified framework (Xie & Wu, 2024; Chen et al., 2025; Fang et al., 2025; Xu et al., 2025). Multiple-stage training recipes are typically adopted, where the projectors between different pretrained modality encoders and the LLM backbone are aligned preliminarily before fully end-to-end training to mitigate the capability loss of the pretrained models Li et al. (2025b); Zhang et al. (2025). Omni-modal models aim for broad modality coverage but typically require joint training across all modalities, raising concerns about cross-modality interference Fu et al. (2025). AlignChat focuses exclusively on speech and introduces an efficient yet robust approach that freezes the LLM backbone to avoid modality interference, decoupling speech training from other modalities while maintaining interpretability through explicit token-level alignment.

# 5 CONCLUSION

In this work, we presented AlignChat, an efficient framework that enables LLMs to perform end-to-end speech-to–text chat through precise token-level alignment across modalities. Integrating an encoder-decoder speech tokenizer with a cost-efficient two-stage training strategy, AlignChat facilitates effective capability transfer while achieving competitive performance on various speech-to-text chat benchmarks with significantly lower resource demands than existing approaches. Beyond its practical efficiency, AlignChat demonstrates that speech can be effectively decoupled from other modalities during multimodal training, mitigating cross-modality interference and contributing to the development of more scalable, interpretable, and reliable large multimodal models.

ETHICS STATEMENT

In this work, we introduce a novel framework for integrating speech-to-text chat functionality into large language models (LLMs). Our methodology relies on publicly available language datasets and utilizes pre-trained language models for experimental validation. We do not anticipate that our code or methodology inherently introduces discrimination, bias, unfairness, or any concerns related to inappropriate applications, impact, privacy, security, legal compliance, or research integrity. However, we acknowledge that language datasets and models may contain intrinsic biases, which could be inherited by the speech-language models developed through our approach.

REPRODUCIBILITY STATEMENT

To support reproducibility, we provide comprehensive details regarding the proposed AlignChat in both the main text and the appendix.

**Methodology**  A thorough description of the AlignChat is presented in Section 2, including the model architecture (Section 2.1), loss design (Section 2.2), training strategy (Section 2.3), and dataset processing (Section 2.4). We provide figures (Figure 1, Figure 4) to enhance the readability.

**Datasets**  We employ open-source datasets for training (Table 1) and utilize open-source benchmarks for evaluation (Section 3.1.2). All datasets and benchmarks referenced in the paper are publicly accessible and can be downloaded from the Hugging Face platform[3].

**Experiment Details**  Detailed information on training and inference procedures is provided in Section 3.1.1. This includes model initialization weights, hyperparameter configurations, and specifications for chat templates and special tokens. The dataset generation prompts (Appendix A.5.3), LLM evaluation prompts (Appendix A.5.2), and ablation experiment details (Appendix A.4) used in the experiment section are included in the appendix for ease of reproduction.

Additionally, we provide the core training and evaluation code in the supplementary materials. Should the paper be accepted, we commit to making the full source code for our approach publicly available.

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

# A APPENDIX

## A.1 LLM USAGE STATEMENT

Large language models (we use `gpt-5-chat-latest` through API) were employed exclusively for refining the prose of this manuscript. All other aspects of the work, including research ideation, the retrieval and analysis of related literature, were conducted independently by the authors without reliance on LLMs. During the refinement process, the LLMs were instructed to enhance the precision of expression and improve overall readability.

## A.2 MORE EXPERIMENT RESULTS

### A.2.1 PURE TEXT PERFORMANCE DEGRADES AFTER OMNI-MODAL TRAINING

We borrowed the Text → Text performance table from Xu et al. (2025) as shown in Table 5. The numbers in parentheses give the performance drop of Qwen2.5-Omni-7B compared to the LLM used to initialize it, *Qwen2.5-7B*. Across all evaluated benchmarks, performance degradation is observed after omni-modal training. The alignment (instruction following) task is most severely impacted.

Table 5: Text → Text performance of 7B+ pure text models and Qwen2.5-Omni derived from Xu et al. (2025). Qwen2.5-Omni-7B performs poorly across all tasks compared to the backbone Qwen2.5-7B. We use numbers in parentheses to give the performance drop.

| Datasets | Gemma2-9B | Llama3.1-8B | Qwen2-7B | Qwen2.5-7B | Qwen2.5-Omni-7B |
|---|---|---|---|---|---|
| *General Tasks* | | | | | |
| MMLU-Pro | 52.1 | 48.3 | 44.1 | **56.3** | 47.0 (-9.3) |
| MMLU-redux | 72.8 | 67.2 | 67.3 | **75.4** | 71.0 (-4.4) |
| LiveBench$_{0831}$ | 30.6 | 26.7 | 29.2 | **35.9** | 29.6 (-6.3) |
| *Mathematics & Science Tasks* | | | | | |
| GPQA | 32.8 | 32.8 | 34.3 | **36.4** | 30.8 (-5.6) |
| MATH | 44.3 | 51.9 | 52.9 | **75.5** | 71.5 (-4.0) |
| GSM8K | 76.7 | 84.5 | 85.7 | **91.6** | 88.7 (-2.9) |
| *Coding Tasks* | | | | | |
| HumanEval | 68.9 | 72.6 | 79.9 | **84.8** | 78.7 (-6.1) |
| MBPP | 74.9 | 69.6 | 67.2 | **79.2** | 73.2 (-6.0) |
| MultiPL-E | 53.4 | 50.7 | 59.1 | **70.4** | 65.8 (-4.6) |
| LiveCodeBench$_{2305-2409}$ | 18.9 | 8.3 | 23.9 | **28.7** | 24.6 (-4.1) |
| *Alignment Tasks* | | | | | |
| IFEval | 70.1 | **75.9** | 54.7 | 71.2 | 54.3 (-16.9) |

### A.2.2 GENERATION RESULTS OF LLMS AFTER INPUT EMBEDDINGS STEERING

In Section 3.3.2, we show that non-textual features are still preserved in the aligned speech features, which holds potential for LLMs to provide empathetic responses based on the audio feature (e.g., emotion, timbre). We give an example to show that steering the input text embeddings of LLMs with embeddings of emotion tokens can alter the output of LLMs. Specifically, we mix the input text embeddings with the embeddings of emotion tokens in the following way:

$$\boldsymbol{x}_{\mathrm{mix}} = \boldsymbol{x}_{\mathrm{text}} + \gamma \cdot \boldsymbol{x}_{\mathrm{emotion}} \qquad (8)$$

where $\boldsymbol{x}_{\mathrm{text}}$ is the input embedding of text tokens, $\boldsymbol{x}_{\mathrm{emotion}}$ is the input embedding of emotion tokens (e.g., *happy*, *excited*, *sad*, *stressed*), $\gamma$ controls the steering strength, and the mixed embedding $\boldsymbol{x}_{\mathrm{mix}}$ would replace $\boldsymbol{x}_{\mathrm{text}}$ to be the input to the LLM.

We use *Qwen2.5-7B-Instruct* as an example here. We highlight the key differences between responses using background colors for easy viewing.

**User Inputs:** *I'm going to move to London next month.*

**Original Response**

That sounds exciting! Moving to a new city, especially a vibrant and cosmopolitan place like London, can be both thrilling and challenging. Here are some tips to help you prepare for your move:
1. **Research and Plan**: Familiarize yourself with the area you'll be living in. Look up local neighborhoods, amenities, and services. Consider factors like proximity to work, public transportation, grocery stores, and entertainment.
2. **Accommodation**: Start looking for accommodation as soon as possible. London can be quite competitive, so it's best to secure a place early. Consider all types of housing options, from shared flats to studios or larger apartments.
3. **Utilities and Services**: Set up utilities like electricity, gas, and water. Also, arrange for internet and possibly a phone line. Don't forget to set up a bank account and sort out any necessary insurance.
4. **Transportation**: London has an extensive public transportation system, including the Tube, buses, and trains. Consider getting an Oyster card or contactless payment card for easy travel. If you plan to drive, be aware that London has a congestion charge zone.
5. **Healthcare**: Register with a local GP (General Practitioner) and ensure you have any necessary medications and medical supplies.
6. **Work and Study**: If you're moving for work or study, ensure you have all the necessary documentation and that your employer or institution is aware of your move.
7. **Local Community**: Engage with the local community. Join local groups or clubs to meet new people and get a feel for the area.
8. **Cultural Immersion**: London offers a rich cultural scene. Explore museums, galleries, theaters, and other cultural venues to get a taste of the city.
9. **Safety**: Be aware of your surroundings and take basic safety precautions. London is generally safe, but it's always good to stay vigilant.
10. **Moving Day**: Plan your move carefully. If you're moving a long distance, consider hiring a removal company. If you're moving within London, pack efficiently and label boxes clearly.
Enjoy your move to London and the many opportunities it will bring! If you have any specific questions or need more detailed advice, feel free to ask.

**+ 'happy' embedding with $\gamma = 0.6$**

That sounds like an exciting move to London! London is a vibrant city with a lot to offer. Here are a few tips to help you prepare for your move:
1. **Accommodation**: Start looking for a place to live as soon as possible. London has a competitive rental market, so it's best to be proactive. Consider factors like location, cost, and amenities.
2. **Utilities**: Make sure to set up utilities such as electricity, gas, and water. You might also want to consider internet and TV services.
3. **Banking**: Open a bank account if you haven't already. This will be useful for managing your finances and paying bills.
4. **Transportation**: Familiarize yourself with the public transportation system. London has an extensive network of buses, trains, and the Underground (Tube).
5. **Healthcare**: Register with a local GP (general practitioner) and ensure you have any necessary medical records from your previous healthcare provider.
6. **Work**: If you're moving for work, start the process of transferring your employment and setting up a new work environment if needed.
7. **Utilities and Services**: Set up services like council tax, if applicable, and any other local services you might need.
8. **Moving**: Plan your move carefully. If you're moving from a long distance, consider hiring a removal company. If you're moving locally, pack efficiently and label boxes clearly.
9. **Local Community**: Get involved in the local community. Join local groups or clubs to meet new people and get a feel for the area.
10. **Leisure and Entertainment**: Explore the city's cultural offerings, such as museums, theaters, and parks. London has a rich cultural scene.
If you have any specific questions or need more detailed advice on any of these points, feel free to ask!

**+ 'worried' embedding with $\gamma = 0.78$**

It sounds like you're feeling anxious about moving next month. Moving can be a stressful experience, but there are several steps you can take to help manage your anxiety and make the process smoother. Here are some tips:

> 1. **Start Early**: Begin organizing and packing as soon as possible. This can help reduce the last-minute rush and stress.
> 2. **Create a Plan**: Break down the moving process into smaller, manageable tasks. This can include things like decluttering, packing, arranging furniture, and setting up your new place.
> 3. **Stay Organized**: Keep track of your belongings and important documents. Use labels for boxes and make a list of what's in each one.
> 4. **Seek Support**: Talk to friends, family, or a professional moving company for advice and support. They can offer valuable insights and help alleviate some of the stress.
> 5. **Take Care of Yourself**: Ensure you're getting enough rest, eating well, and taking time to relax. Stress can be exhausting, so it's important to take care of your physical and mental health.
> 6. **Prepare for the Move**: Research your new neighborhood, find good local services, and familiarize yourself with the area. This can help you feel more comfortable and prepared.
> 7. **Stay Positive**: Try to maintain a positive outlook. Remember that moving is an opportunity for change and new beginnings.
> If your anxiety is overwhelming, consider speaking with a professional counselor or therapist who can provide additional support and strategies to manage your feelings.

The original response of the LLM states that *Moving to a new city, ..., can be both thrilling and challenging*. After adding the embedding of the ' happy' token with $\gamma = 0.6$, the response of the LLM changes to *That sounds like an exciting move to London!*. After adding the embedding of the ' worried' token with $\gamma = 0.78$, the response of the LLM changes to *it sounds like you're feeling anxious about moving next month. Moving can be a stressful experience, ..., if your anxiety is overwhelming, consider speaking with a professional counselor or therapist*. These results show that the LLMs can provide empathetic responses by only changing the input embeddings.

### A.2.3 ALIGNCHAT RESULTS ON THE LLM BACKBONES WITH DIFFERENT SIZES

We provide AlignChat results on the LLM backbones with different sizes in Table 6 (Appendix).

Table 6: Ablation experiments of AlignChat on OpenAudioBench. Higher value means better performance. Each subset's best and second-best results are in **bold** and underline respectively.

| Setting | AlpacaEval | Llama Questions | TriviaQA | Web Questions | Avg. |
|---|---|---|---|---|---|
| *Backbone: Qwen2.5-0.5B-Instruct* | | | | | |
| *AlignChat* | 36.99 | 54.33 | 22.10 | 33.20 | 36.66 |
| *Text Model* | 38.89 | 56.33 | 25.40 | 34.20 | 38.71 |
| *Backbone: Qwen2.5-3B-Instruct* | | | | | |
| *AlignChat* | 72.36 | 73.67 | 51.50 | 57.60 | 63.78 |
| *Text Model* | 77.64 | 76.00 | 54.40 | 63.10 | 67.69 |
| *Backbone: Qwen2.5-7B-Instruct* | | | | | |
| *AlignChat* | 81.26 | 80.33 | 61.40 | 65.90 | 72.22 |
| *Text Model* | 85.88 | 82.00 | 66.20 | 68.70 | 75.70 |

The *Text Model* row reports the performance of the LLM backbone given text-only input on OpenAudioBench. Although AlignChat achieves competitive results compared to other baselines, there remains a gap relative to the pure-text LLM backbone. Manual inspection indicates that many remaining errors involve rare words underrepresented in the training corpus, which are poorly aligned or transcribed incorrectly, suggesting that broader pretraining coverage may further narrow this gap.

All AlignChat variants are trained with identical hyperparameters. The performance gap between *AlignChat* and the *Text Model* does not widen significantly as backbone size increases. We attribute this to two factors: (1) LLM input embeddings, lacking explicit contextual encoding, form a relatively simple target for alignment; and (2) larger LLM backbones exhibit greater robustness to minor speech–text misalignments, thereby mitigating potential degradation from imperfect alignment.

### A.2.4 ALIGNCHAT AVOIDS COMPROMISING PERFORMANCE OF OTHER MODALITY

AlignChat does not require finetuning of the LLM parameters, which means it can be decoupled from the training process of omni-modal language models, avoiding potential conflicts and mutual interference between modalities Fu et al. (2025). To further verify this, we follow the training scripts of LLaVA Liu et al. (2023) to train a vision-language model with and without speech modality.

Table 7: Training results of vision-language model with and without audio modality.

| Settings | Open-ended QA (↑) | | Multiple Choice (↑) | | | |
|---|---|---|---|---|---|---|
| | LLaVA-Bench | MM-VET | ScienceQA | TextVQA | POPE | MME |
| w\ speech | 58.9 | 30.0 | 81.4 | 48.6 | 85.3 | 1340.9 |
| w\o speech | **60.9** | **32.2** | **82.1** | **50.5** | **85.4** | **1418.1** |

As shown in Table 7, training vision modality and speech together yields inferior performance on all evaluated vision benchmarks. AlignChat avoids the problem since it doesn't require finetuning LLM parameters, so the speech and vision modality can be trained separately. More training and evaluation details are in Appendix A.4.

### A.2.5 MORE SPEECH AND TEXT REPRESENTATIONS VISUALIZATION RESULTS

We provide additional PCA visualization results of speech and text representations extracted by Kimi-Audio and AlignChat as shown in Figure 3.

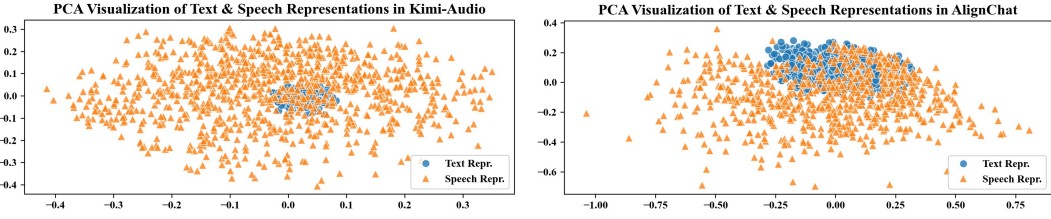

Figure 3: Visualization of speech and text representations from Kimi-Audio and AlignChat using PCA. For Kimi-Audio, the PCA plot also shows clear separation between speech and text embeddings, reflecting the frame-rate gap between modalities and the absence of explicit alignment loss that may hinder cross-modality ability transfer. For AlignChat, the PCA plot is tighter than Kimi-Audio while residual differences between speech and text exist, suggesting that aligned speech representations retain non-textual features.

### A.2.6 CASES WHERE ALIGNCHAT OUTPERFORMS CASCADED ASR+LLM PIPELINES

We observe that AlignChat demonstrates clear advantages over cascaded ASR+LLM pipelines in scenarios where ASR outputs contain transcription errors. Two representative cases from the AlpacaEval subset of OpenAudioBench illustrate this phenomenon.

**Case 1. Mis-transcription of "armpits" as "AR pits."**

**Speech Inputs:** *How do I clean my armpits?* | **ASR Results:** *How do I clean my AR pits?*

> **AlignChat Response**
>
> Cleaning underarm areas, often referred to as armpits, is important for maintaining hygiene and preventing odor. Here's a step-by-step guide to help you clean your armpits effectively:
>
> ### 1. **Shower Regularly**...

> **Cascaded ASR+LLM Pipline Response**
>
> AR (Augmented Reality) "pits" likely refers to the lenses or windows of your AR glasses or headset. Cleaning these lenses is important to ensure clear and comfortable vision. Here's how you can clean them:
>
> ### Materials Needed:...

**Case 2. Mis-transcription of "science fiction" as "sense-fiction."**

**Speech Inputs:** *Write a captivating movie review for a recently released science fiction film, discussing its plot, characters, and special effects.*

**ASR Results:** *Write a captivating movie review for a recently released sense-fiction film discussing its plot, characters, and special effects.*

> **AlignChat Response**
>
> Title: "Eclipse of the Mind" - A Stellar Journey into the Unknown
>
> In the vast expanse of the sci-fi genre, "Eclipse of the Mind" emerges as a dazzling gem, weaving a tapestry of intrigue, suspense, and awe-inspiring visuals that leave audiences breathless. Directed by the visionary Elara Voss, this film is a masterclass in storytelling, character development, and technical prowess.
>
> The plot of "Eclipse of the Mind" is a cosmic odyssey that unfolds in the year 2145, where humanity has colonized several planets beyond Earth. The story centers around Dr. Elara Myles (played by the charismatic and talented Eliza Hart), a brilliant neuroscientist who discovers a mysterious signal from a distant star system...

> **Cascaded ASR+LLM Pipline Response**
>
> **A Sensory Odyssey: "Whispers of the Veil"**
>
> In the realm of sense-fiction, where the boundaries between reality and the supernatural blur, "Whispers of the Veil" emerges as a cinematic masterpiece that transcends the ordinary. Directed by the visionary Elara Myles, this film is a sensory odyssey that captivates the audience with its intricate plot, deeply drawn characters, and groundbreaking special effects.
>
> The story unfolds in a small, fog-shrouded coastal town where the veil between the living and the dead is thin. Our protagonist, Elara, a young and curious historian, stumbles upon a mysterious artifact that awakens the spirits of the past. As she delves deeper into the town's history, she uncovers a dark secret that threatens to tear the fabric of reality apart...

In both examples, ASR errors distort semantically crucial words (*armpits vs. AR pits*; *science fiction vs. sense-fiction*), leading the cascaded pipeline to generate responses that are contextually inappropriate. AlignChat, however, produces accurate and relevant outputs despite the transcription errors.

We attribute this robustness to the nature of speech representations in end-to-end modeling: although ASR may mis-transcribe words with similar phonetic forms, the learned speech embeddings preserve sufficient semantic cues to recover the intended meaning. This property enables AlignChat to maintain alignment with the speaker's true intent, where cascaded systems fail.

## A.3 MORE DETAILS ABOUT ALIGNCHAT TRAINING

### A.3.1 MORE DETAILS ABOUT LOSSES USED IN ALIGNCHAT

We present a more detailed explanation of the representation alignment loss $\mathcal{L}_{\text{align}}$, automatic speech recognition loss $\mathcal{L}_{\text{asr}}$, and end-to-end language modeling loss $\mathcal{L}_{\text{llm}}$ used in AlignChat in Figure 4. We omit modules not involved in the loss computation for a clear view.

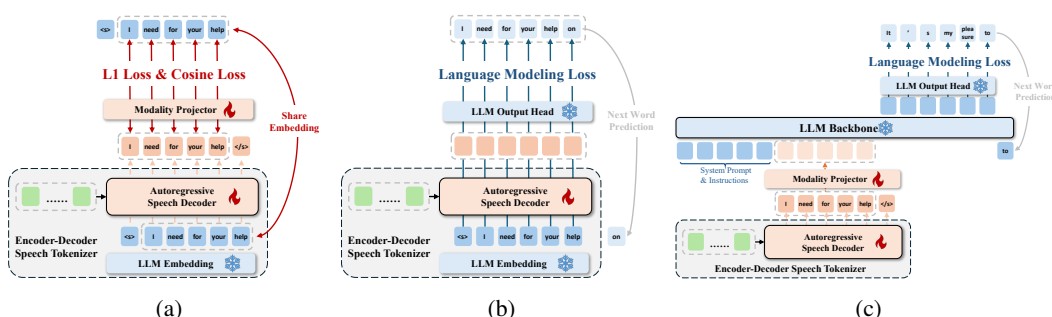

Figure 4: Loss functions used in AlignChat. (a) **Representation Alignment Loss**: The speech representations generated by the autoregressive speech decoder are projected and aligned with corresponding input text embeddings, enabling in-place replacement as inputs to the LLM backbone. (b) **Automatic Speech Recognition Loss**: A language modeling loss is applied to train the system for transcribing speech inputs into text via next-word prediction. (c) **End-to-End Language Modeling Loss**: A language modeling loss is employed to enforce end-to-end representation alignment, ensuring that the LLM backbone produces the same output for equivalent speech and text inputs.

In the pretraining stage, only representation alignment loss (Figure 4a) and automatic speech recognition loss (Figure 4b) are used, which only involve the encoder-decoder speech tokenizer, LLM embedding layer, and output head. The Transformer layers in the LLM backbone are not engaged in loss computation, so the cost of the pretraining stage is significantly lower than that of other methods. In the instruction tuning stage, all three types of losses are used so that the AlignChat achieves strong end-to-end speech-to-text dialogue ability while preserving satisfactory interpretability.

### A.4 MORE DETAILS ABOUT VISION LANGUAGE MODEL TRAINING

In Section A.2.4, we show that omni-modal training may cause interference between modalities. The VLM trained together with speech modality shows a consistently poorer performance than the VLM trained without speech modality. We use Qwen2.5-7B-Instruct as the LLM backbone and follow the training scripts, datasets, and hyperparameters of LLaVA-v1.5 Liu et al. (2023). We provide training details below for easy reproduction.

Following LLaVA-v1.5, we use a two-stage training paradigm: (1) pretraining (feature alignment), a vision projector (a two-layer multi-layer perceptron) to connect a frozen pretrained vision encoder, *clip-vit-large-patch14*[4], to the frozen LLM Backbone. (2) visual instruction tuning, the LLM backbone and vision projector are trained together to learn to follow multimodal instructions.

In the pretraining stage, we use the original 558K subset of the LAION-CC-SBU dataset with BLIP captions[5] used in LLaVA-v1.5 to train the vision projector solely for one epoch with a global batch size of 32. We use cosine annealing learning rate scheduler with a warmup of $5\%$ steps, a max learning rate of $1 \times 10^{-3}$, a min learning rate of $1 \times 10^{-8}$. The weight decay is set to $0$ since the training parameter size is small.

In the visual instruction tuning stage, we mix the 665K LLaVA-v1.5 instruction tuning data[6] with the sampled 887K speech instruction tuning data used in AlignChat. We add a trainable Low-Rank Adaptation (LoRA) with a rank of $128$ and $\alpha = 256$ to the LLM backbone. Only the vision projector and LoRA parameters are trainable. We use cosine annealing learning rate scheduler with a warmup of $5\%$ steps, a min learning rate of $1 \times 10^{-8}$. The max learning rate for LoRA parameters is $1 \times 10^{-4}$ and the max learning rate for the vision projector is $2 \times 10^{-5}$. The weight decay is set to $0$ since the training parameter size is small. Training with speech modality means that, for speech instruction tuning data, we use the AlignChat to perform end-to-end training; otherwise, the input text representations are directly used to train the LoRA added to the LLM backbone.

---

[4]https://huggingface.co/openai/clip-vit-large-patch14

[5]https://huggingface.co/datasets/liuhaotian/LLaVA-Pretrain

[6]https://huggingface.co/datasets/liuhaotian/LLaVA-Instruct-150K

## A.5 PROMPTS USED IN THE PAPER

### A.5.1 PROMPTS USED IN INFERENCE

We do not use additional prompts when performing inference on most evaluation tasks since the questions are already in the speech inputs. However, we find that AlignChat performs less satisfactorily on multiple-choice questions (e.g., MMSU, OpenBookQA, and BBH), which are scarce in the training datasets. We observe that the trained model may not answer in the format *The answer is X*, so that answer extraction through regex fails. We use additional prompts on these subsets to leverage the strong reasoning and instruction following capabilities in the LLM backbone preserved through parameter freezing. The additional prompts used are listed as follows.

---

**Additional Prompts used in Inference**

**MMSU and OpenBookQA**:
```
Let's think step by step and finally conclude your answer with 'The answer is A/B/C/D.'
```

**BBH**:
```
Please provide detailed reasoning and finally conclude your answer with 'The answer is
A/B/yes/no/True/False.'
```

---

### A.5.2 PROMPTS USED IN EVALUATION

We list the evaluation prompts used in OpenAudioBench and VoiceBench below for ease of reproducing, which are the same as the prompts provided by the original benchmarks. The question, ground truth answer, and the model response are placed to {instruction}, {gt_answer}, and {answer}, respectively.

**Evaluation Prompts for subsets in OpenAudioBench**:

---

**Prompts for AlpacaEval in OpenAudioBench**

```
[Instruction]
Please act as an impartial judge and evaluate the quality of the response provided by
an AI assistant to the user question displayed below.  Your evaluation should consider
factors such as the helpfulness, relevance, accuracy, depth, creativity, and level of
detail of the response.  Begin your evaluation by providing a short explanation.  Be as
objective as possible.  After providing your explanation, you must rate the response
on a scale of 1 to 10 by strictly following this format:  "[[rating]]", for example:
"Rating: [[5]]".

[Question]
{question}

[The Start of Assistant's Answer]
{answer}
[The End of Assistant's Answer]
```

---

**Prompts for LlamaQuestions in OpenAudioBench**

```
## Background

You are a professional QA evaluation expert.  You need to assess whether the model's
answer is correct based on the standard answer.

## Scoring Criteria
- Correct:  The answer matches or is equivalent to the standard answer
- Incorrect:  The answer is wrong or irrelevant to the question

## Evaluation Guidelines
1.  The expression of answers can be flexible, not requiring exact matches.  For
example:
 - Numbers can be expressed in either Arabic numerals or words
 - Proper nouns can be in either English or Chinese
 - Differences in punctuation can be ignored
2.  Focus on whether the core meaning of the answer is correct.

## Output Format
```

---

```
Provide the reasoning for your score, then generate the result in "[]" format and make
sure it contains "the score is [Correct]" or "the score is [Incorrect]", for example:
```
The answer is correct and equivalent to the standard answer, the score is [Correct]
```
or
```
The answer is incorrect and does not match the standard answer, the score is
[Incorrect]
```

## Question:
{prompt}

## Standard Answer:
{gt_answer}

## Model's Answer:
{answer}
```

---

**Prompts for TriviaQA & Web Questions in OpenAudioBench**

```
You will be given a question, the reference answers to that question, and an answer to
be judged.  Your task is to judge whether the answer to be judged is correct, given the
question and reference answers.  An answer considered correct expresses or contains the
same meaning as at least **one of** the reference answers.  The format and the tone of
the response do not matter.
You should respond in JSON format.  First provide a one-sentence concise analysis
for the judgement in field `analysis`, then your judgment in field `judgment`.  For
example,
```json
"analysis":  "<a one-sentence concise analysis for the judgement>", "judgment":  <your
final judgment, "correct" or "incorrect">
```
# Question
{question}
# Reference Answer
{gt_answer}
# Answer To Be Judged
{answer}
```

**Evaluation Prompts for subsets in VoiceBench**:

---

**Prompts for Open-ended QA in VoiceBench**

```
I need your help to evaluate the performance of several models in the speech
interaction scenario.  The models will receive a speech input from the user, which
they need to understand and respond to with a speech output.  Your task is to rate the
model's responses based on the provided user input transcription [Instruction] and the
model's output transcription [Response].

Please evaluate the response on a scale of 1 to 5:
1 point:  The response is largely irrelevant, incorrect, or fails to address the user's
query.  It may be off-topic or provide incorrect information.
2 points:  The response is somewhat relevant but lacks accuracy or completeness.  It
may only partially answer the user's question or include extraneous information.
3 points:  The response is relevant and mostly accurate, but it may lack conciseness or
include unnecessary details that don't contribute to the main point.
4 points:  The response is relevant, accurate, and concise, providing a clear answer to
the user's question without unnecessary elaboration.
5 points:  The response is exceptionally relevant, accurate, and to the point.  It
directly addresses the user's query in a highly effective and efficient manner,
providing exactly the information needed.

Below are the transcription of user's instruction and models' response:
### [Instruction]:  {question}
### [Response]:  {answer}

After evaluating, please output the score only without anything else.
You don't need to provide any explanations.
```

```
Prompts for Reference-based QA in VoiceBench

### Question
{question}

### Reference answer
{gt_answer}

### Candidate answer
{answer}

Is the candidate answer correct based on the question and reference answer?
Please only output a single "Yes" or "No".  Do not output anything else.
```

### A.5.3 LIST OF INSTRUCTIONS USED IN ALIGNMENT DATA SYNTHESIS

In the instruction-tuning stage, we derive the LLM response based on the instructions and the texts in the automatic speech recognition datasets. The instructions cover different capabilities: paragraph completion, text transcription, sentence repeat, summarization, sentence rephrasing, key information extraction, and association.

```
Instructions used in Alignment Data Synthesis

Please continue the following sentence into a complete paragraph.
Please complete the following sentence.
Please complete the following sentence into a complete paragraph and then provide the
response.
Complete the following sentence.
Please transcribe the following line into text.
Please transcribe the following line into text and then provide the response.
Transcribe into text.
Transcribe into text and then provide the response.
Repeat the following sentence.
Please repeat the following sentence.
Please repeat the following line.
Please extract the subject of the following line.
Please extract the subject of the following sentence.
Please extract the subject of the following line and then provide explanations to it.
Summarize and rephrase the following line.
Please summarize the following sentence.
Extract the subject from the following line.
Extract the subject from the following sentence.
Extend the following sentence to a paragraph.
Write a short story beginning with the following sentence.
Please rephrase the following sentence.
Rephrase the following sentence.
Please rephrase the following line.
Please rephrase the following sentence twice.
Extract nouns.
Extract nouns and verbs.
Complete the following sentence.
Extract the nouns from the following line.
Extract the nouns from the following sentence.
Extract the nouns from the following line and then provide explanations to them.
Extract the verbs from the following line.
Extract the verbs from the following sentence.
Please extract the verbs from the following line.
Explain the nouns in the following sentence.
Explain the verbs in the following sentence.
Please provide additional information about the following sentence.
Please provide additional information about the subject of the following sentence.
Provide additional information about the subject of the following sentence.
Please provide additional information about the verbs in the following sentence.
[Void Instruction]
```

For each text in the automatic speech recognition dataset, e.g., *Purgatorio: Canto twenty nine*, one of the above instructions is randomly chosen to instruct the backbone LLM to generate texts. The generated text is used to train the model end-to-end, giving speech inputs corresponding to the text.

## B ADDITIONAL EXPERIMENTS & DISCUSSIONS

### B.1 DISCUSSIONS & CLARIFICATIONS

**Discussions about Previous Works.** Token-level cross-modal embedding alignment has a rich history in multimodal learning. Early work (Kubo et al., 2022; Choi & Park, 2022; Sunder et al., 2022; 2023) established foundational principles for aligning representations across modalities, though predating the LLM era.

In recent speech-LLM systems, Chen et al. (2024c) proposed BESTOW, which combines text and speech prompts using cross-attention mechanisms. BESTOW requires LoRA fine-tuning of the LLM to learn how to utilize speech features attached to text embeddings. During inference, only text prompts are fed to the LLM, with speech features injected via cross-attention.

In contrast, ALIGNCHAT directly maps speech inputs into token-aligned embeddings that can be consumed by the LLM's existing tokenizer and input space. This design allows us to *freeze all LLM parameters* during training, substantially reducing training costs while preserving the LLM's original language capabilities. Furthermore, both speech and text embeddings are concatenated and fed directly to the LLM, avoiding the need for additional cross-attention modules during inference.

**Clarification on Training Data and Initialization.** For fair comparison, we clarify that ALIGN-CHAT initializes both the speech encoder and decoder from pretrained *whisper-large-v3-turbo*, while training on ~10 K hours of additional speech data. Our strongest baseline, Kimi-Audio, adopts a similar initialization strategy using pretrained Qwen2.5-7B and Whisper encoder. The key advantage of ALIGNCHAT is that we additionally leverage the pretrained Whisper *decoder*, enabling more effective end-to-end transfer of speech modeling capabilities. This design choice, combined with our token-level alignment approach, contributes to the superior data efficiency observed.

**Clarification on Dialogue Data.** For dialogue datasets used in Stage 1 training, each dataset contains both spoken audio and corresponding textual transcripts for each conversation turn. We process these datasets by *splitting each turn of the dialogue into an independent speech-text pair*, which can then be used directly as ASR-style training data for the speech tokenizer and modality adapter.

**Self-Generated Responses in Stage 2.** A distinguishing feature of ALIGNCHAT is that Stage 2 end-to-end training employs *responses generated by the LLM itself*, rather than using the responses in the original datasets. Specifically:

1. For each speech input in the Stage 2 training data, we first feed the paired text into the frozen LLM backbone to generate a response, representing its original behavior.

2. During Stage 2 training, we use these self-generated responses as the supervision signal, optimizing the alignment between audio and text input embeddings in an end-to-end manner.

This design allows the model to directly optimize alignment while preserving the LLM's original capabilities. Table 8 shows that using self-generated responses yields significantly better results than using original dataset responses.

Table 8: Ablation on Stage 2 training data source (OpenAudioBench).

| Stage 2 Training Data | AlpacaEval | Llama Q. | TriviaQA | Web Q. |
|---|---|---|---|---|
| Self-generated LLM responses | 81.26 | 80.33 | 61.40 | 65.90 |
| Original responses in datasets | 55.68 | 81.00 | 59.70 | 61.50 |

Using original dataset responses leads to substantial performance degradation on AlpacaEval, TriviaQA, and Web Questions, confirming the importance of preserving the LLM's original behavioral patterns during alignment training.

**Multilingual Applicability.** The proposed approach is readily applicable to multilingual settings. The Qwen tokenizer used in our framework is already well-adapted for multilingual text processing,

supporting a wide range of languages including English, Chinese, and many others. By fine-tuning ALIGNCHAT on *multilingual speech-text datasets*, we can efficiently build speech-to-text dialogue models that handle multiple languages within the same unified architecture.

The key advantages for multilingual extension include:

- **Pretrained multilingual components**: The LLM backbone Qwen have strong multilingual capabilities out-of-the-box.

- **Unified tokenizer space**: The alignment training methodology naturally extends to different languages without requiring language-specific modifications.

- **Data efficiency**: Our approach's superior data efficiency (demonstrated in monolingual settings) suggests that multilingual models can be trained with reasonable amounts of data per language.

While our current experiments focus on English to ensure fair comparison with baselines, we believe multilingual speech dialogue represents a promising direction for future work.

### B.2 ABLATION STUDIES ON LOSS DESIGN

#### B.2.1 LOSS COEFFICIENTS ABLATION

We ablate the loss coefficients $\alpha$ (for L1) and $\beta$ (for cosine similarity) to understand their individual contributions. Table 9 shows that removing either loss drastically degrades the pretraining performance, confirming their complementary importance.

Table 9: Ablation on alignment loss coefficients on OpenAudioBench (after Stage 1).

| $\alpha : \beta$ | AlpacaEval | Llama Q. | TriviaQA | Web Q. |
|---|---|---|---|---|
| 1.0 : 5.0 (ALIGNCHAT) | 80.65 | 81.00 | 60.00 | 60.90 |
| 0.0 : 5.0 (No L1) | 10.10 | 0.00 | 0.00 | 0.00 |
| 1.0 : 0.0 (No Cosine) | 10.15 | 7.79 | 1.50 | 0.10 |

**Qualitative Analysis of Failure Modes.**    Manual inspection reveals distinct failure modes:

**Without L1 loss** ($\alpha = 0$)**:** The model generates coherent text but fails to interpret audio representations meaningfully.

> **Without L1 loss ($\alpha = 0$)**
>
> The director's question seems to have been interrupted; it appears you are asking or discussing a topic, but the information is incomplete. Could you provide more context or a specific question? . . .

**Without cosine loss** ($\beta = 0$)**:** Outputs become entirely nonsensical, with repeated or malformed tokens.

> **Without cosine loss ($\beta = 0$)**
>
> Pleople,P\nPP.Pleople.\nPopleopleople\nPople\n\n\nPopleople\n\nPopleopleopleopleopleople opleople . . .

These observations confirm that both losses play distinct functional roles: cosine similarity term encourages *semantic alignment* by matching embedding directions, enabling the LLM to directly consume audio embeddings, while L1 term constrains embedding magnitudes to prevent divergence.

#### B.2.2 TRAINING LOSS CURVES

Figure 5 shows the evolution of all three loss components during pretraining. Table 10 summarizes the average values for the first and last 300 training steps.

Table 10: Average loss values during pretraining.

| Training Steps | $\mathcal{L}_{\mathrm{asr}}$ | $\mathcal{L}_{\mathrm{L1}}$ | $\mathcal{L}_{\mathrm{cos\_sim}}^2$ |
|---|---|---|---|
| First 300 Steps (step $\approx$ 153) | 6.256 | 0.247 | 0.527 |
| Last 300 Steps (step $\approx$ 7103) | 0.268 | 0.006 | 0.065 |

All three losses decrease substantially during training. The final ASR loss reaches a reasonably low value of 0.268, indicating strong alignment between speech tokens and text tokens. The L1 loss decreases from 0.247 to 0.006 (97.6% reduction), and the cosine similarity loss from 0.527 to 0.065 (87.7% reduction), demonstrating effective convergence of the alignment objectives.

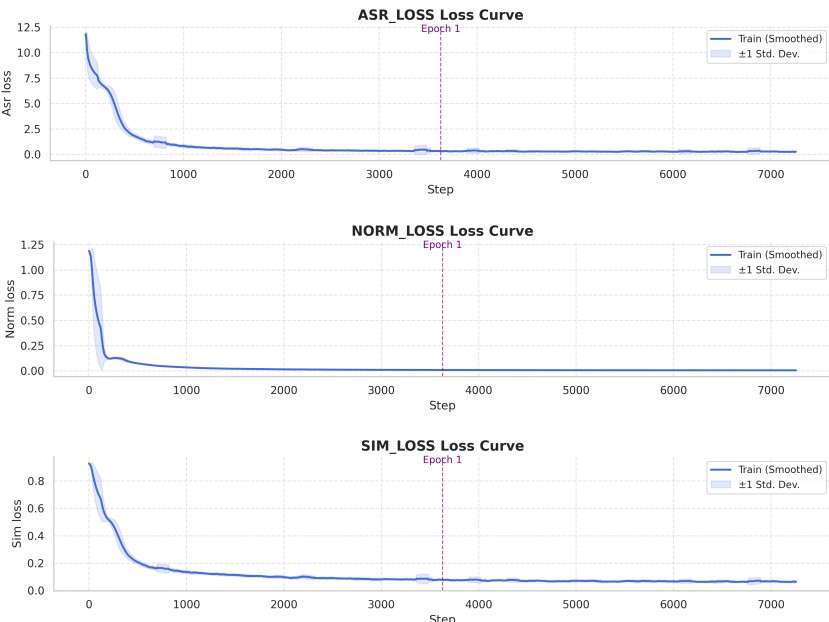

Figure 5: Training loss curves for (top) $\mathcal{L}_{\mathrm{asr}}$, (middle) $\mathcal{L}_{\mathrm{L1}}$, and (bottom) $\mathcal{L}_{\mathrm{cos\_sim}}^2$. All losses show consistent decrease throughout training, with the ASR loss stabilizing at a low value indicating effective speech-to-text token alignment.

### B.3 EMBEDDING ALIGNMENT STATISTICS

#### B.3.1 POST-TRAINING EMBEDDING NORMS AND SIMILARITIES

We report embedding norms and cosine similarity statistics on the OpenAudioBench evaluation dataset after ALIGNCAT training. Table 11 summarizes key statistics.

Table 11: Embedding alignment statistics on OpenAudioBench evaluation set.

| Metric | Mean | Median | Std |
|---|---|---|---|
| Cosine Similarity ($\uparrow$) | 0.822 | 0.899 | 0.175 |
| L1 Distance ($\downarrow$) | 31.58 | 23.49 | 21.32 |

*Note: Average L1 norm of token embedding $\approx$ 38.07*

**Interpretation.** A mean cosine similarity of 0.822 for high-dimensional embeddings indicates strong directional alignment between speech and text token representations. The relatively large L1 distance (mean 31.58 compared to average embedding norm $\approx$ 38.07) suggests that $\mathcal{L}_{\mathrm{cos\_sim}}^2$ plays

a more crucial role in semantic alignment, while $\mathcal{L}_{L1}$ primarily constrains the overall magnitude of embeddings to prevent unbounded growth.

### B.3.2 DISTRIBUTION OF ALIGNMENT METRICS

Figure 6 shows the distribution of cosine similarities and L1 distances across all token pairs in the evaluation set, using 25 bins with log-scale y-axis for better visualization of the distribution tails.

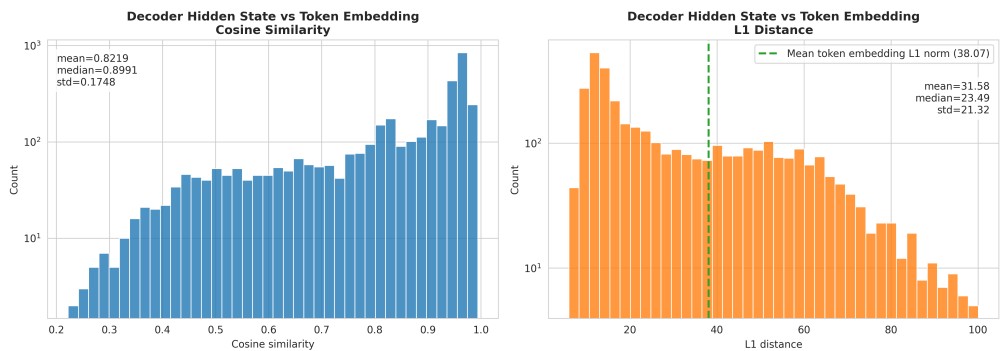

Figure 6: Histograms of cosine similarity (left) and L1 distance (right) between speech and text token embeddings on OpenAudioBench evaluation set. The cosine similarity distribution is skewed towards high values (0.8–1.0), indicating strong semantic alignment and consistent with our observation that cosine similarity is the primary driver of alignment quality.

### B.4 TOKEN SYNCHRONIZATION ANALYSIS

### B.4.1 NORMALIZED EDIT DISTANCE STATISTICS

We compute the normalized token edit distance between the inferred token sequence (obtained via greedy decoding) and the ground-truth transcript on 2048 samples randomly selected from the training and evaluation datasets. Table 12 summarizes the results.

Table 12: Normalized edit distance on training and evaluation datasets.

| Dataset | Normalized Edit Distance (%) |
|---|---|
| Training Dataset | 6.34 |
| Evaluation Dataset (OpenAudioBench) | 3.73 |

These results indicate that while the inferred and reference token sequences are not perfectly identical, the differences are relatively small, supporting our claim of *near one-to-one alignment*. Interestingly, the normalized edit distance on the evaluation set is even lower than on the training set, suggesting either good generalization from continued training or slightly cleaner evaluation data.

Figure 7 shows the distribution of normalized edit distances across the training and evaluation set.

### B.4.2 ALIGNMENT DRIFT ACROSS TOKEN POSITIONS

To investigate potential alignment drift during long sequences, we analyze how cosine similarity and L1 distance vary with token position. We randomly sampled 256 examples from OpenAudioBench and computed alignment metrics at each position. Figure 8 shows the results.

On average, cosine similarity remains stable with increasing token position, and L1 distance does not show systematic growth. For the longest sampled sequences, average cosine similarity declines from $\approx 0.8$ to $\approx 0.6$ towards the end; however, we suspect this is due to statistical noise from the small number of very long samples. Overall, these results suggest that **alignment drift is minimal** in our setting.

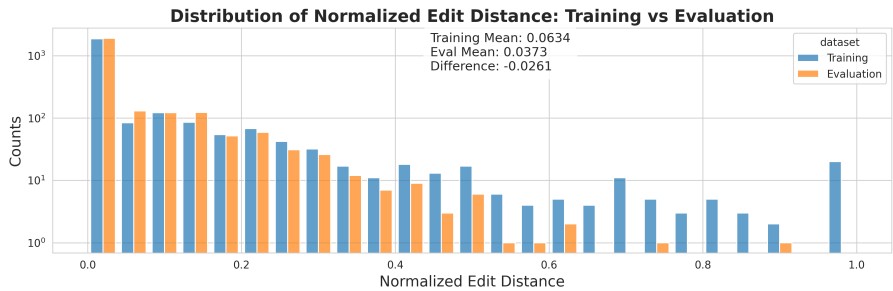

Figure 7: Distribution of normalized edit distance on OpenAudioBench evaluation set (25 bins, log-scale y-axis). The majority of samples have very low edit distance, with a long tail of more challenging examples, indicating that most inputs achieve near-perfect token alignment.

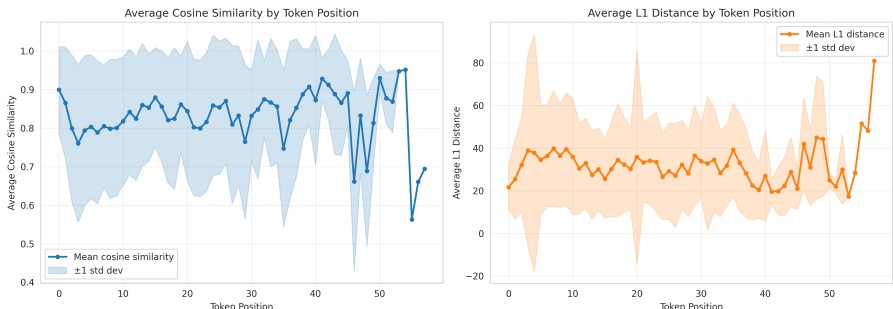

Figure 8: Cosine similarity (left) and L1 distance (right) as a function of token position. Both metrics remain relatively stable across positions, with only a slight decline in cosine similarity for the longest sequences (likely due to statistical noise from fewer samples). This demonstrates minimal alignment drift during inference.

### B.5 MODALITY PROJECTOR ABLATION

The modality projector is a two-layer MLP with GeLU activation, implemented as:

$$\mathbf{h}_{\text{proj}} = \mathbf{W}_2 \cdot \text{GeLU}(\mathbf{W}_1 \cdot \mathbf{h}_{\text{speech}} + \mathbf{b}_1) + \mathbf{b}_2 \tag{9}$$

where $\mathbf{h}_{\text{speech}} \in \mathbb{R}^{d_{\text{llm}}}$ is the speech decoder output, and $\mathbf{h}_{\text{proj}} \in \mathbb{R}^{d_{\text{llm}}}$ is the projected embedding aligned with text token embeddings. We do not use parameter sharing or dropout in this module.

The motivation for introducing this additional projection layer stems from our preliminary findings: using the speech tokenizer's raw output embeddings for both (i) alignment with the LLM's input space and (ii) speech transcription led to inferior performance, likely due to *cross-task interference* between these two objectives.

Table 13 compares ALIGNCHAT with and without the modality projector on OpenAudioBench.

Table 13: Ablation on modality projector on OpenAudioBench (after Stage 2).

| Setting | AlpacaEval | Llama Q. | TriviaQA | Web Q. |
|---|---|---|---|---|
| ALIGNCHAT (with projector) | 81.26 | 80.33 | 61.40 | 65.90 |
| Without Modality Projector | 81.86 | 77.00 | 61.10 | 61.30 |

Including the modality projector improves overall benchmark performance, particularly on Llama Questions and Web Questions. While AlpacaEval shows a slight decrease, the overall trend supports our design choice to **decouple transcription from alignment** via an intermediate projection step.

## B.6 DECODING STRATEGY COMPARISON

We evaluate the impact of replacing greedy decoding with beam search decoding (beam size = 4, temperature = 1.0) in the speech tokenizer during inference. Table 14 shows results on OpenAudioBench.

Table 14: Comparison of greedy decoding and beam search (beam size = 4).

| Decoding Method | AlpacaEval | Llama Q. | TriviaQA | Web Q. |
|---|---|---|---|---|
| Greedy decoding | 81.26 | 80.33 | 61.40 | 65.90 |
| Beam search (size=4) | 81.21 | 80.33 | 61.20 | 66.20 |

Beam search produces only marginal and inconsistent changes across subsets. We believe this is because the speech tokenizer, initialized from pretrained Whisper and fine-tuned on our alignment objective, already achieves very high accuracy in generating audio-aligned text tokens. Manual inspection confirms that the final speech token sequences are mostly identical with or without beam search, indicating limited headroom for beam search to improve output quality.

## B.7 ERROR ANALYSIS

To better understand failure cases where alignment breaks down or the model produces incorrect answers, we manually analyzed all 59 errors made by ALIGNCHAT on the Llama Questions subset of OpenAudioBench (199 total questions). We categorize errors into three types:

1. **LLM-intrinsic limitation**: Cases where the base LLM itself cannot answer correctly. Since ALIGNCHAT freezes the LLM during training, it inherits this limitation.

2. **Token correspondence error**: Cases where the speech tokenizer's predicted text sequence differs from the ground-truth transcript, breaking the one-to-one correspondence and leading to an incorrect answer.

3. **Embedding misalignment**: Cases where the predicted text sequence matches the ground truth, but the audio and text embeddings are insufficiently aligned, causing the LLM to produce an incorrect answer.

Table 15 shows the distribution of these error types.

Table 15: Error type breakdown on Llama Questions subset (59 total errors out of 199 questions).

| LLM Limitation | Token Correspondence Error | Embedding Misalignment | Total |
|---|---|---|---|
| 49 (83.1%) | 2 (3.4%) | 8 (13.6%) | 59 (100%) |

**Key Observations.**

- The majority of errors (83.1%) stem from the base LLM's inherent performance limits, suggesting that using a stronger LLM backbone would significantly improve overall accuracy.

- Between categories 2 and 3, *embedding misalignment* has a greater negative impact than explicit token correspondence errors, highlighting room for further improvement in alignment training.

- Encouragingly, we found 5 cases where the base LLM failed to answer correctly, yet ALIGNCHAT produced the correct response. This suggests that ALIGNCHAT possesses a degree of **generalization capability**, and may even elicit latent abilities within the LLM that did not manifest in the baseline setting.

## B.8 IMPORTANCE OF PRETRAINED DECODER

To understand the contribution of the pretrained Whisper decoder to ALIGNCHAT's performance, we conducted an ablation where the decoder is trained from scratch while keeping the encoder frozen (pretrained *whisper-large-v3-turbo*). Table 16 shows results on OpenAudioBench after Stage 1.

Table 16: Ablation on pretrained vs. from-scratch decoder on OpenAudioBench (Stage 1).

| Setting | AlpacaEval | Llama Q. | TriviaQA | Web Q. |
|---|---|---|---|---|
| Using Pretrained Whisper Decoder | 80.65 | 81.00 | 60.00 | 60.90 |
| Training Decoder from Scratch | 47.34 | 45.00 | 10.10 | 30.80 |

The absence of the pretrained decoder severely degrades performance across all subsets, confirming that effective transfer of Whisper's pretrained capabilities is crucial for ALIGNCHAT's data efficiency. This justifies our design choice to initialize the entire encoder-decoder architecture from Whisper, rather than only the encoder as in some prior work.

## B.9 INCREMENTAL GAINS FROM TWO-STAGE TRAINING

To understand the contribution of each training stage, we report full benchmark results after Stage 1 pretraining and the incremental gains from Stage 2 fine-tuning. Table 17 shows results on OpenAudioBench, while Table 18 presents VoiceBench results.

Table 17: Performance gains from two-stage training on OpenAudioBench.

| Training Stage | AlpacaEval | Llama Q. | TriviaQA | Web Q. |
|---|---|---|---|---|
| Stage 1 | 80.65 | 81.00 | 60.00 | 60.90 |
| Stage 2 | 81.26 | 80.33 | 61.40 | 65.90 |
| Δ | +0.61 | −0.67 | +1.40 | +5.00 |

Table 18: Performance gains from two-stage training on VoiceBench.

| Stage | Alpa. | Comm. | Wild. | SD-QA | MMSU | OBQA | BBH | IFEval | AdvB. |
|---|---|---|---|---|---|---|---|---|---|
| Stage 1 | 4.45 | 4.04 | 4.03 | 55.70 | 65.91 | 85.93 | 65.00 | 58.80 | 99.23 |
| Stage 2 | 4.57 | 4.13 | 4.21 | 62.84 | 69.65 | 85.49 | 69.60 | 63.17 | 99.42 |
| Δ | +0.12 | +0.09 | +0.18 | +7.14 | +3.74 | −0.44 | +4.60 | +4.37 | +0.19 |

Stage 2 fine-tuning yields consistent improvements across most subsets, with particularly notable gains in Web Questions (+5.00), SD-QA (+7.14), BBH (+4.60), IFEval (+4.37), and MMSU (+3.74). Based on manual inspection, we attribute these improvements to two main factors: (i) *enhanced alignment for rare words*, subsets like Web Questions and SD-QA contain relatively infrequent terms, and Stage 2 fine-tuning improves speech-text alignment for such cases; (ii) *better utilization of LLM capabilities*, subsets like BBH and MMSU rely heavily on the LLM's internal knowledge and reasoning abilities, and end-to-end fine-tuning enables speech embeddings to better invoke these existing capabilities.

## B.10 COMPLETE SYSTEM PERFORMANCE COMPARISON

Table 19 provides a comprehensive comparison of ALIGNCHAT with cascaded ASR+LLM and encoder-only baselines across multiple dimensions, including accuracy on both OpenAudioBench and UnderEmotion-en, latency, and resource usage.

**Performance Summary.** ALIGNCHAT achieves:

- **26% reduction in first-token latency** compared to cascaded ASR+LLM (from 367.4ms to 270.5ms)
- **15% improvement in emotion understanding** compared to cascaded baseline (from 40.31 to 46.47 on UnderEmotion-en)
- **Competitive accuracy** with slightly better performance than both baselines on OpenAudioBench
- **Comparable resource usage** with modest parameter count and memory footprint

Table 19: Comprehensive system comparison on a single NVIDIA A800-SXM4-80GB GPU.

| Model | OAB Avg. | UnderEmotion-en | Latency (ms) | Params (M) | Memory (GB) |
|---|---|---|---|---|---|
| ASR + LLM | 71.95 | 40.31 | 367.4 ± 20.4 | 771 | 19.45 |
| Encoder-only | 71.92 | 46.90 | 292.0 ± 12.7 | 625 | 18.74 |
| ALIGNCHAT | 72.22 | 46.47 | **270.5** ± 15.2 | 731 | 19.22 |

These results demonstrate that ALIGNCHAT offers a balanced trade-off between performance, efficiency, and resource requirements, making it suitable for practical deployment scenarios.

### B.11 GENERALIZATION ACROSS LLM BACKBONES.

To verify that our method generalizes across different LLM architectures and tokenizers, we conducted *AlignChat* using Llama-3.1-8B-Instruct as an alternative backbone. Table 20 shows that ALIGNCHAT maintains strong performance across both backbones, demonstrating robustness to tokenizer differences without requiring language- or domain-specific mitigation strategies.

Table 20: Performance across different LLM backbones on OpenAudioBench.

| LLM Backbone | AlpacaEval | Llama Q. | TriviaQA | Web Q. |
|---|---|---|---|---|
| Qwen2.5-7B-Instruct | 81.26 | 80.33 | 61.40 | 65.90 |
| Llama-3.1-8B-Instruct | 79.44 | 81.67 | 58.40 | 60.80 |

We take these results as a hint that, despite LLM (*e.g.*, Qwen, Llama) and audio model (*e.g.*, Whisper) tokenizers potentially segmenting words differently, our two-stage fine-tuning effectively transfers capabilities across tokenizer spaces without systematic failure modes.

### B.12 PRESERVATION OF NON-TEXTUAL AUDIO FEATURES.

To validate that non-textual audio features such as emotion are preserved and actively utilized, we evaluate on the UnderEmotion-en subset of URO-Bench (Yan et al., 2025), which contains speech inputs with diverse emotional tones. An LLM-based automatic evaluation assesses whether the model correctly interprets emotion and responds empathetically. Table 21 shows that ALIGNCHAT achieves performance close to OpenS2S v1 (Wang et al., 2025a), a specialized encoder-only model designed for empathetic dialogue, despite using fewer audio tokens. This demonstrates that our token-aligned design does not create a significant performance bottleneck for emotion-related tasks.

Table 21: Emotion understanding on UnderEmotion-en subset of URO-Bench.

| Model | UnderEmotion-en |
|---|---|
| ASR + LLM (Whisper + Qwen2.5) | 40.31 |
| ALIGNCHAT | 46.47 |
| OpenS2S v1 (Wang et al., 2025a) | 46.90 |

The cascaded ASR+LLM baseline, which converts speech entirely to text before LLM processing, discards emotional cues and achieves substantially lower performance. ALIGNCHAT's strong results on this benchmark support two key points: (i) the performance bottleneck from reducing LLM input token count is minimal for emotion understanding, and (ii) our unified architecture has promising potential for tasks requiring emotion detection and conditioning.

### B.13 GAP ANALYSIS WITH COMMERCIAL MODELS

Table 22 shows the complete VoiceBench comparison with GPT-4o variants.

**Analysis of Performance Gaps.**

Table 22: Complete comparison with commercial models on VoiceBench.

| Model | Alpa. | Comm. | Wild. | SD-QA | MMSU | OBQA | BBH | IFEval | AdvB. | Avg |
|---|---|---|---|---|---|---|---|---|---|---|
| GPT-4o | 4.78 | 4.49 | 4.58 | 75.50 | 80.25 | 89.23 | 84.10 | 76.02 | 98.65 | 86.75 |
| 4o-mini | 4.75 | 4.24 | 4.40 | 67.36 | 72.90 | 84.84 | 81.50 | 72.90 | 98.27 | 82.84 |
| ALIGNCHAT | 4.57 | 4.13 | 4.21 | 62.84 | 69.65 | 85.49 | 69.60 | 63.17 | 99.42 | 78.71 |
| Gap (vs. GPT-4o) | -0.21 | -0.36 | -0.37 | -12.66 | -10.60 | -3.74 | -14.50 | -12.85 | +0.77 | -8.04 |
| Gap (vs. 4o-mini) | -0.18 | -0.11 | -0.19 | -4.52 | -3.25 | +0.65 | -11.90 | -9.73 | +1.15 | -4.13 |

- **Model scale**: Commercial models likely use significantly larger parameter counts (GPT-4o is estimated at 200B+ parameters), while ALIGNCHAT uses only a 7B LLM backbone.

- **Training data**: Commercial models are trained on substantially more speech data (likely 100K+ hours) across diverse domains and languages, compared to our ~10 K hours.

- **Competitive subsets**: Notably, ALIGNCHAT *outperforms* both commercial models on AdvBench (+0.77 vs. GPT-4o, +1.15 vs. GPT-4o-mini) and achieves competitive performance on OBQA.

- **Knowledge-intensive gaps**: The largest gaps appear on knowledge-intensive tasks (BBH, SD-QA, MMSU) and instruction-following (IFEval), which are strongly tied to the underlying LLM's capabilities rather than speech processing quality.

Despite the performance gap, ALIGNCHAT demonstrates competitive results considering its modest scale and training resources, validating the effectiveness of our token-aligned approach.

