# OpenReview forum: "AlignChat: Endowing LLMs with End-to-End Speech-to-Text Chat Capability through Token-Level Representation Alignment"
_ICLR.cc/2026/Conference — Submitted to ICLR 2026_

### Official Review · Reviewer_8EeD · 2025-10-18

**Soundness:** 2
**Presentation:** 2
**Contribution:** 2
**Rating:** 2
**Confidence:** 5

**Summary:**

The authors try to address the resolution mismatch issue among speech and text token representation in speech LLM. They propose to use AED based framework to synchronize the speech token with text token. Two stages training is employed with ASR pre-training and instruct fine-tuning. The results show good performance on 3 benchmarks.
My main concern is novelty. The proposed method is quite similar to BESTOW [1] but with a more powerful LLM and better results on three speech benchmarks. Also, the comparison in this paper is unfair. The proposed system can only generate text modality but most systems compared in this work could generate both text and speech outputs.

[1] Chen, Zhehuai et al. “Bestow: Efficient and Streamable Speech Language Model with The Best of Two Worlds in GPT and T5.” SLT (2024)

**Strengths:**

The authors build a good system with the proposed method, which shows good performance on 3 open-source benchmarks.

**Weaknesses:**

- The authors consider the resolution mismatch of speech token and text token could hurt the performance and propose to use AED based framework to integrate speech token into the LLM. However, the work doesn't give direct evidence to support this claim. It would be great if authors could conduct similar comparison similar as [1] to give more evidence. Also, the ablation study in table 3 also shows that extra decoder only provides small gain compared with the AED framework.
- The proposed method is quite similar to BESTOW and no discussion is provided in the related work section.
- The proposed system is not able to generate audio compared with other baseline systems in Table 2.


[1] Lam, Tsz Kin et al. “Prepending or Cross-Attention for Speech-to-Text? An Empirical Comparison.” NAACL (2025)

**Questions:**

- How does this method compared with a fully cascaded approach, i.e., ASR (Whisper)  and LLM (Qwen2.5-7B-Instruct)?
- What's the impact of modality projector given the alignment and ASR training on the decoder?

---

> ### Author Response · Authors · 2025-11-24
>
> **Part 1/2**
>
> ---
>
> We sincerely thank the reviewer for taking the time to review our paper. We appreciate that you found that we build a good system with the proposed method, which shows good performance on open-source benchmarks.
>
> ---
>
> **Weakness 1 & Question 1**: Clarify the Strength of AlignChat
>
> > The authors consider the resolution mismatch of speech token and text token could hurt the performance and propose to use AED based framework to integrate speech token into the LLM. However, the work doesn't give direct evidence to support this claim. It would be great if authors could conduct similar comparison similar as [1] to give more evidence. Also, the ablation study in table 3 also shows that extra decoder only provides small gain compared with the AED framework.
> >
> > [1] Lam, Tsz Kin et al. “Prepending or Cross-Attention for Speech-to-Text? An Empirical Comparison.” NAACL (2025)
>
> >  How does this method compared with a fully cascaded approach, i.e., ASR (Whisper) and LLM (Qwen2.5-7B-Instruct)?
>
> **Response to Weakness 1**: We thank the reviewer for noting the need for direct evidence supporting AlignChat's advantage. Following your suggestion, we provide a **comprehensive comparison** between AlignChat, an encoder‑only end‑to‑end method, and a strong cascaded ASR + LLM pipeline, measured on accuracy, first‑token latency, parameter count, and GPU memory usage (single NVIDIA A800 GPU):
>
> | Model         | OpenAudioBench Avg. | First‑Token Latency (ms) | Parameters (M) | GPU Memory (GB) |
> | ------------- | ------------------- | ------------------------ | -------------- | --------------- |
> | ASR + LLM      | 71.95               | $367.4 \pm 20.4$       | 771            | 19.45           |
> | Encoder‑only  | 71.92               | $292.0 \pm 12.7$       | 625            | 18.74           |
> | AlignChat     | 72.22               | $270.5 \pm 15.2$       | 731            | 19.22           |
>
> Although AlignChat achieves only a modest gain in accuracy on the current test set, we view it as a **balanced trade‑off** between encoder‑only and cascaded architectures:
>
> 1. **Versus cascaded ASR + LLM**: AlignChat offers **true end‑to‑end capability**, removing the intermediate transcription step and enabling efficient parallel execution of the audio and text modules.
> 2. **Versus encoder‑only**: AlignChat uses **fewer audio tokens** and requires lower training cost, yet achieves superior performance on *speech‑centric tasks*. As LLM backbone size increases, encoder‑only methods suffer more from increased inference latency and computational cost, whereas AlignChat's pretraining involves only the LLM's embedding layers, making it a **cost‑efficient choice**.
>
> The cascaded ASR + LLM pipeline remains a strong baseline, but prior work and our results highlight its inherent limitations:
> - **Tokenizer mismatch & lack of joint optimization**: Different tokenizers for ASR/LLM block end‑to‑end optimization and hinder parallelization, impacting latency metrics.
> - **Loss of fine‑grained audio cues**: Text conversion discards prosody, timbre, emotion, and other subtle features, limiting empathetic response capability.
>
> *Evidence of preserved audio information.*
> To validate our claim that AlignChat retains non‑textual audio features, we evaluated it on the **UnderEmotion‑en** subset of URO‑Bench [1], using automatic LLM‑based assessment of emotional understanding in dialogue responses:
>
> | Model         | UnderEmotion‑en |
> | ------------- | --------------- |
> | ASR + LLM      | 40.31           |
> | AlignChat     | 46.47           |
> | OpenS2S v1 [2] | 46.90           |
>
> Here, the cascaded ASR + LLM baseline lacks explicit emotional cues, while OpenS2S v1 is a specialised encoder‑only empathetic dialogue model. **AlignChat achieves performance close to OpenS2S v1**, reinforcing that:
> - Reducing LLM input token count in speech tasks does **not cause significant performance bottlenecks**, and
> - AlignChat can **extend effectively to tasks requiring richer audio information**, such as empathetic dialogue generation.

---

> ### Author Response · Authors · 2025-11-24
>
> **Part 2/2**
>
> ---
>
> **Weakness 2**: Comparison with BESTOW
>
> > The proposed method is quite similar to BESTOW and no discussion is provided in the related work section.
>
> **Response to Weakness 2**: We thank the reviewer for pointing out the highly relevant BESTOW work. While the overall architectures share certain similarities, **AlignChat** and **BESTOW** differ substantially in **motivation** and **inference strategy**.
>
> BESTOW is designed to effectively combine **text prompts** and **speech prompts**, using a Transformer module with **cross‑attention** so that text prompts can selectively extract task‑relevant information from speech prompts during generation. In contrast, AlignChat is motivated by the goal of **directly converting speech input into embeddings that can be reused by the LLM's existing tokenizer and input space**, avoiding the need for an intermediate cross‑attention fusion stage.
>
> For example, in a speech translation task:
> - **BESTOW** feeds the LLM with only the text prompt (e.g., *“Please translate the speech from English to French”*), and injects speech features via cross‑attention to augment this prompt. This design requires additional **LoRA fine‑tuning of the LLM** so that it learns to use speech features attached to text embeddings.
> - **AlignChat**, on the other hand, first maps the speech input into token‑aligned textual embeddings, concatenates these embeddings with the text prompt, and feeds them directly into the LLM. This allows AlignChat to **leverage the original language abilities of the pretrained LLM**, keeping **all LLM parameters frozen** during training and substantially lowering training cost.
>
> We will add a **Related Works** discussion in the revised manuscript explicitly comparing AlignChat and BESTOW, clarifying both the architectural parallels and the key distinctions in design goals and usage.
>
> ---
>
> **Weakness 3**: Inable to Generate Audio
>
> > The proposed system is not able to generate audio compared with other baseline systems in Table 2.
>
> **Response to Weakness 3**: We thank the reviewer for raising this important point. AlignChat is currently focused on **speech‑to‑text dialogue capabilities** and does not generate audio responses.
>
> We have ensured fairness in our comparisons in two ways:
> 1. **Consistent evaluation setting**: All models, including those capable of speech generation, are evaluated in a speech‑to‑text configuration, rather than comparing speech‑output accuracy.
> 2. **Comparable data accounting**: We report only the ASR training data hours for other models, excluding speech‑generation training data, to avoid inflating their reported resource usage.
>
> While it is possible that training a model to generate speech could affect its dialogue performance, AlignChat's advantage remains clear given its **significantly lower training data requirements** compared to baselines. Furthermore, we believe AlignChat's **token‑level audio‑text alignment design** can be extended to speech generation, making audio output a promising area for future development.
>
> ---
>
> **Question 2**: Reason why using Modality Projector
>
> > What's the impact of modality projector given the alignment and ASR training on the decoder?
>
> **Response to Question 2**: We thank the reviewer for noting the need for clarification. In preliminary experiments, we found that using the **speech tokenizer's raw output embeddings** for both alignment with LLM inputs **and** speech transcription led to **inferior performance**, likely due to **cross‑task interference**.
>
> To address this, we introduced an additional **modality projector**: the speech tokenizer's output is first passed through this projector, and the resulting embeddings are aligned with the LLM's text‑token embeddings.
>
> *Ablation results on OpenAudioBench:*
>
> | Setting                   | AlpacaEval | Llama Questions | TriviaQA | Web Questions |
> | ------------------------- | ---------- | --------------- | -------- | ------------- |
> | **AlignChat**             | 81.26      | 80.33            | 61.40    | 65.90         |
> | **Without Modality Projector** | 81.86      | 77.00            | 61.10    | 61.30         |
>
> Including the modality projector improves **overall benchmark performance**. This supports our design choice to explicitly **decouple transcription from alignment**.
>
> ---
>
> > **Reference**:
> >
> > [1] URO-Bench: Towards Comprehensive Evaluation for End-to-End Spoken Dialogue Models.
> >
> > [2] OpenS2S: Advancing Fully Open-Source End-to-End Empathetic Large Speech Language Model.

---

> ### Comment · Reviewer_8EeD · 2025-11-25
> **Comments on the feedback**
>
> Thanks authors for providing detailed feedback.
>
> First, I am not sure about encoder only system in the comparison experiments newly added. Are you referring to a CTC or Transducer based system? Or it is a typo and you are referring to the decoder only system. I would assume the latter.
> Based on the results, I still not convinced that the proposed system is better than the cascaded system or the decoder only system.
> Also, why the decoder only system has larger latency compared with the AlignChat system, though the decoder only system is with less parameters?
>
> `In preliminary experiments, we found that using the speech tokenizer's raw output embeddings for both alignment with LLM inputs and speech transcription led to inferior performance, likely due to cross‑task interference.`
> Any explanation about this? Why the native LLM embedding is worse?

---

> > ### Author Response · Authors · 2025-11-25
> >
> > **Question 1**
> >
> > > First, I am not sure about encoder only system in the comparison experiments newly added. Are you referring to a CTC or Transducer based system? Or it is a typo and you are referring to the decoder only system. I would assume the latter. Based on the results, I still not convinced that the proposed system is better than the cascaded system or the decoder only system. Also, why the decoder only system has larger latency compared with the AlignChat system, though the decoder only system is with less parameters?
> >
> > **Response to Question 1**: We thank the reviewer for raising this question and would like to clarify a terminology issue first. In our paper, *encoder‑only* refers to speech–language models where the speech encoder directly produces embeddings for the LLM input **without an intermediate speech decoder**, which is a default practice for the baseline approaches, e.g., *Kimi‑Audio*, *Qwen2.5‑Omni*, *SLAM-Omni*, and *LLaMA‑Omni*. We are **not** referring to a decoder‑only architecture here.
> >
> > A typical encoder‑only pipeline proceeds as follows:
> > - The input audio is processed by a pretrained **Whisper encoder** to produce high‑frequency (~50 Hz) output features.
> > - These features are down‑sampled to a lower frame rate before being fed into the LLM (e.g., Kimi‑Audio downsamples to 12.5 Hz).
> > - Even after down‑sampling, the resulting sequence length is still **significantly longer** than the number of tokens produced by AlignChat’s speech decoder (~3 Hz).
> >
> > Because the LLM must process **more audio tokens per query** in encoder‑only systems, **first‑token latency can be higher**, despite the smaller parameter count, due to the quadratic (or super‑linear) scaling of transformer attention with sequence length.
> >
> > AlignChat adopts an **encoder–decoder speech tokenizer** that outputs token sequences **aligned one‑to‑one with text tokens**. This keeps the LLM input shorter, enabling **faster inference** while preserving rich audio features. This architectural choice explains why AlignChat achieves lower latency than encoder‑only systems in our benchmarks, in addition to its advantages over the cascaded ASR + LLM baseline.
> >
> > We hope this clarifies the encoder‑only terminology, the methodological approach (with references to Kimi‑Audio, Qwen2.5‑Omni, and LLaMA‑Omni), and how sequence length differences underlie the observed latency patterns.
> >
> > ---
> >
> > **Question 2**
> >
> > > `In preliminary experiments, we found that using the speech tokenizer's raw output embeddings for both alignment with LLM inputs and speech transcription led to inferior performance, likely due to cross‑task interference.` Any explanation about this? Why the native LLM embedding is worse?
> >
> > **Response to Question 2**: We thank the reviewer for these important questions and for giving us the opportunity to clarify our experimental setup and findings.
> >
> > *Clarifying the "without modality projector" setting.*
> > The ablation experiment comparing “with” vs. “without” modality projector **does not** involve replacing or evaluating the LLM’s native text‑token embeddings.
> > In the *without modality projector* setting, the **speech tokenizer’s raw decoder output embeddings** are used **both** for:
> >
> > 1. ASR transcription, and
> > 2. Direct alignment with the LLM’s text‑token embedding space for downstream language modeling.
> >
> > This means the same embedding stream must simultaneously satisfy **two distinct optimization objectives**:
> >
> > - **ASR transcription** - preserving fine‑grained acoustic detail (phonetics, prosody, speaker variation) for accurate token‑level recognition.
> > - **LLM semantic alignment** - matching the distribution and semantic characteristics of the pretrained LLM’s input embedding space, where detailed acoustic distinctions are largely irrelevant.
> >
> > We also stress that we do not use the LLM’s native text embeddings as a substitute for speech embeddings, because **doing so would be equivalent to a cascaded ASR + LLM pipeline** — this *discards all non‑textual audio information* (e.g., emotion, timbre, prosody). In our earlier evaluation of the UnderEmotion-en subset of URO-Bench, such pipelines perform significantly worse (40.31 for ASR + LLM) than AlignChat (46.47), precisely because *emotional cues are lost in pure text transcriptions*.
> >
> > *How the modality projector helps.*
> > The **modality projector** introduces a lightweight, trainable mapping between the speech decoder and the LLM’s input space, which:
> > - **Decouples** transcription from alignment
> > - Allows the speech decoder to specialize in ASR objectives
> > - Let the projector transform ASR‑optimized embeddings into semantically‑aligned representations for the LLM
> >
> > This separation reduces gradient interference and better matches the representational requirements of each task. The experiment above provides empirical evidence that including the modality projector improves **overall benchmark performance**.
> >
> > ---
> >
> > We would be happy to address any further questions you may have.

---

### Official Review · Reviewer_NsNf · 2025-10-30

**Soundness:** 4
**Presentation:** 3
**Contribution:** 3
**Rating:** 2
**Confidence:** 4

**Summary:**

This paper proposes an alignment framework between acoustic embeddings and corresponding LLM-based token embedding. The authors utilize an attention-based encoder-decoder (AED) ASR model (Whisper) to extract acoustic embeddings from the decoder which is aligned against the LLM embeddings. AlignChat achieves this through a two-stage process: first mapping speech representations into the LLM’s embedding space, then instruction-tuning the model to follow prompts directly from audio. By freezing the LLM and introducing only a lightweight speech adapter, the method preserves the model’s reasoning ability while enabling end-to-end speech interaction. Experiments show that AlignChat performs competitively on speech dialogue benchmarks with a fraction of the training data used by prior systems.

**Strengths:**

Strengths:

1. The paper is well written and easy to follow.
2. The paper tackles an important problem.
3. The problem formulation is expressed clearly and is mathematically sound.
4. Overall impressive results for speech-to-text chat tasks.
5. Comprehensive analysis in the form of ablation studies and visualization of learnt representations.

**Weaknesses:**

Weaknesses:

1. This paper misses out on citing some key initial papers on token-level cross-modal embedding alignment. For example,

[1] Kubo et al., "Knowledge transfer from large-scale pretrained language models to end-to-end speech recognizers"

[2] Choi et al., "Distilling a pretrained language model to a multilingual ASR model"

[3] Sunder et al., "Tokenwise contrastive pretraining for finer speech-to-bert alignment in end-to-end speech-to-intent systems"

[4] Sunder et al., "Fine-Grained Textual Knowledge Transfer to Improve RNN Transducers for Speech Recognition and Understanding"

While these paper were before the LLM era, the underlying philosophy of embedding alignment is exactly the same and hence they merit to be acknowledged.

2. Some missing experiments/discussion about cascaded v/s end-to-end approaches (see next section).
3. Lack of clarity in the design of $L_{align}$ as a combination of $L1$ loss and cosine similarity loss (see next section).
4. Please see next section for other questions and comments.

**Questions:**

1. The fact that the authors used the decoder output of an AED ASR model as the acoustic embedding for alignment and subsequent input to the LLM raises some questions:

a. How do the authors deal with the mismatch during training and inference? During training, ground-truth transcripts are available, therefore, there is indeed a one-to-one correspondence between the acoustic and text embeddings. However, there will always be errors during inference, in particular, insertion and deletion errors which prohibit this alignment. How is this taken into account.

b. The authors suggest that they use greedy decoding during inference from the AED model. How does this compare with beam search decoding apart from the obvious latency differences?

c. How is $L_{speech} = L_{text}$ (line 139) ensured during test time?

d. The authors mention in line 264, "During inference, speech inputs are first encoded into audio features and then autoregressively converted into speech representations that maintain a strict correspondence with text tokens". That cannot be true in the presence of ASR errors. Please re-phrase.

e. Finally, I don't see baseline numbers where Whisper was used as a first-pass ASR model and the output of this is fed to the LLM. In other words, I think it would be useful to have a cascaded baseline. Although the authors present some qualitative results in the appendix, some quantitative results are necessary where one should use a very strong state-of-the-art ASR model and cascade it with a state-of-the-art LLM since this is a widely adopted pipeline in many speech-to-text chat applications.

2. In line 156, "The modality projector decouples transcription from representation alignment to avoid cross-task interference". Does the speech decoder not get adapted with the alignment? If so, how is it decoupled?
3. The authors propose to use $L_{align}$ as a combination of $L1$ loss and cosine similarity loss such that non-textual and textual features can be decoupled. However, the effect or benefit of this is not clear from table 4. Although we see that non-textual features like emotion and gender information is preserved, we do not know if this can be achieved from just a simple $L1$ loss or cosine-similarity loss.
4. From table 1, it seems that the authors use dialogue data for stage 1 training. How is a non-ASR data used for stage 1, since it only involves the AED ASR model and the modality adapter?
5. In line 231, "for each speech input, the paired text is fed into the LLM backbone to generate the response representing its original behavior." At what stage is this data used?

---

> ### Author Response · Authors · 2025-11-24
>
> **Part 1/4**
>
> ---
>
> We sincerely thank the reviewer for taking the time to review our paper. We appreciate that you found our paper is well written and easy to follow, solving an important problem and achieving overall impressive results.
>
> ---
>
> **Weakness 1**: Miss Citing Important Works
>
> > This paper misses out on citing some key initial papers on token-level cross-modal embedding alignment.
>
> **Response to Weakness 1**: We thank the reviewer for highlighting this important oversight. We fully agree that the earlier work on *token‑level cross‑modal embedding alignment*, even though predating the LLM era, is philosophically aligned with our approach and has directly influenced the development of alignment methods across modalities. We will **add citations to the key papers you mentioned** in the revised manuscript.
>
> ---
>
> **Weakness 2 & Question 1e**: Comparison with Cascaded Baseline
>
> > Some missing experiments/discussion about cascaded v/s end-to-end approaches (see next section).
>
> > Finally, I don't see baseline numbers where Whisper was used as a first-pass ASR model and the output of this is fed to the LLM. In other words, I think it would be useful to have a cascaded baseline.
>
> **Response to Weakness 2 & Question 1e**: We thank the reviewer for this insightful question. The key difference between **AlignChat** and a conventional cascaded ASR + LLM pipeline lies in the **interface between the speech module and the LLM**:
>
> - In AlignChat, the speech tokenizer passes **intermediate audio embeddings** directly to the LLM, making it an **end‑to‑end approach**.
> - In the cascaded baseline, the ASR first **transcribes** the audio into text, which is then re‑tokenized and fed into the LLM.
>
> *Inference behavior.*
> During inference, the decoder of the speech tokenizer produces both (i) audio token embeddings for the LLM's end‑to‑end comprehension, and (ii) a predicted text token sequence to maintain an approximate one‑to‑one correspondence between audio and text tokens when no ground‑truth transcript is available.
>
> While cascaded ASR + LLM remains a strong baseline in speech‑dialogue tasks, research has increasingly shifted toward end‑to‑end approaches due to two inherent limitations of cascading:
> 1. **Tokenizer mismatch and lack of joint optimization**: Common ASR systems (e.g., Whisper) and LLMs use different tokenizers. ASR output must be fully transcribed to text and then re‑tokenized for the LLM, preventing direct end‑to‑end optimization and limiting parallelization. This impacts latency‑critical metrics such as first‑token delay.
> 2. **Loss of fine‑grained speech features**: Converting speech to text eliminates prosody, timbre, emotion, and other subtle acoustic cues, hindering the construction of more **empathetic** dialogue systems.
>
> *Quantitative comparison.*
> We benchmarked AlignChat and a cascaded ASR + LLM pipeline on a single NVIDIA A800 GPU:
>
> | Model         | OpenAudioBench Avg. | UnderEmotion‑en | First‑Token Latency (ms) | Parameters (M) | GPU Memory (GB) |
> | ------------- | ------------------- | ---------------- | ------------------------ | -------------- | --------------- |
> | ASR + LLM      | 71.95               | 40.31             | $367.4 \pm 20.4$       | 771            | 19.45           |
> | AlignChat     | 72.22               | 46.47             | $270.5 \pm 15.2$       | 731            | 19.22           |
>
> Here, **OpenAudioBench Avg.** measures overall audio‑to‑text dialogue capability. **UnderEmotion‑en** is a subset of the URO‑Bench [1] dataset focusing on emotional speech inputs, evaluated automatically for empathetic response quality.
>
> AlignChat shows a **clear advantage** over the cascaded baseline in first‑token latency and emotion‑related performance, with a small but consistent gain in overall dialogue accuracy. Beyond noise effects, we identified qualitative cases where AlignChat outperformed due to its ability to leverage richer audio embeddings:
>
> > **Question**: How do I clean my armpits?
> >
> > **ASR Output**: How do I clean my *AR pits*?
> >
> > **ASR + LLM Response**: *AR (Augmented Reality) "pits"* likely refers to the lenses or windows of your AR glasses or headset. Cleaning these lenses is important...
> >
> > **AlignChat Response**: *Cleaning underarm areas, often referred to as armpits*, is important for maintaining hygiene and preventing odor. Here's a step-by-step guide...
>
> In this example, an ASR recognition error propagated directly into the cascaded pipeline's answer, while AlignChat's LLM correctly inferred the intended meaning from the audio embeddings and responded appropriately.
>
> Overall, the end‑to‑end AlignChat framework delivers **lower latency**, **better emotion understanding**, and **greater robustness to ASR errors** compared with cascaded ASR + LLM pipelines, while preserving strong general dialogue performance.

---

> > ### Comment · Reviewer_NsNf · 2025-11-25
> >
> > 1. The authors mention how cascaded ASR+LLM has limitations. While I agree with the limitations, I disagree that "research has increasingly shifted toward end‑to‑end approaches". While there is definitely research going on in the domain of end-to-end approaches, it is not fair to sideline ASR+LLM cascaded approach as a lot modern systems rely on that. See table 3 in [1].
> >
> > 2. The authors present a cascaded baseline in their rebuttal. I appreciate the effort. However, it is not clear which ASR model, or which LLM was used, it is difficult for me to judge this experiment. Ideally, you would want a SOTA ASR and LLM for this. Furthermore, I feel that it will be good to have this baseline in table 2 as an additional row.
> >
> > 3. How does the cascaded model perform on Voicebench and also what is the breakdown of scores on different subset.
> >
> > [1] Chen et al., VoiceBench: Benchmarking LLM-Based Voice Assistants

---

> > > ### Author Response · Authors · 2025-11-26
> > >
> > > **Question 3**: Cascaded Baseline Details and VoiceBench Breakdown
> > >
> > > > ... it is not clear which ASR model, or which LLM was used, it is difficult for me to judge this experiment. Ideally, you would want a SOTA ASR and LLM for this....
> > > >
> > > > How does the cascaded model perform on Voicebench and also what is the breakdown of scores on different subset.
> > >
> > > **Response to Question 3**: We thank the reviewer for their constructive suggestions regarding fair comparison to strong cascaded baselines.
> > >
> > > Our cascaded ASR + LLM baseline follows reviewer XoS6’s recommendation: we use **whisper‑large‑v3‑turbo** fine‑tuned on our data as the ASR model and **Qwen2.5‑7B‑Instruct** as the LLM backbone for fair comparison with AlignChat, which is initialized from the same ASR model and LLM backbone.
> > >
> > > We now provide results on **VoiceBench** together with several common commercial multi‑modal systems:
> > >
> > > | Model                    | AlpacaE. | CommonE. | WildVoice | SD‑QA  | MMSU   | OBQA   | BBH    | IFEval | AdvBench | Avg.   |
> > > |--------------------------|----------|----------|----------|--------|--------|--------|--------|--------|----------|--------|
> > > | Whisper‑v3‑large + GPT‑4o | 4.80     | 4.47     | 4.62     | 75.77  | 81.69  | 92.97  | 87.20  | 76.51  | 98.27    | 87.80  |
> > > GPT-4o-Audio                  |    4.78    |    4.49    |   4.58    | 75.50 | 80.25 | 89.23 | 84.10 | 76.02  |  98.65   |  86.75  |
> > > | **ASR + LLM**            | 4.71     | 4.41     | 4.66     | 70.25  | 74.27  | 88.13  | 76.40  | 71.03  | 98.84    | 83.84  |
> > > | **AlignChat**            | 4.57     | 4.13     | 4.21     | 62.84  | 69.65  | 85.49  | 69.60  | 63.17  | 99.42    | 78.71  |
> > >
> > > *Comparison with ASR + LLM baseline.*
> > > The ASR + LLM baseline achieves stronger overall scores than AlignChat on VoiceBench, especially on *knowledge/reasoning‑intensive subsets* (e.g., SD‑QA, MMSU, BBH) and *instruction‑following tasks* (IFEval). However, **AlignChat already achieves strong performance among open-source end-to-end speech dialogue models**. We see considerable room for improvement by combining our alignment architecture with enhanced alignment strategies, allowing AlignChat to close the gap across tasks while *retaining its advantage in emotion‑sensitive and latency‑critical scenarios*.
> > >
> > > *Comparison with Commercial Models.*
> > > AlignChat does not yet match the performance of proprietary commercial speech‑dialogue models, but still achieves competitive scores given (i) its *modest 7B parameter size*, and (ii) use of only *~10K hours of speech data* for training. The largest gaps appear in **knowledge/reasoning‑intensive tasks** (BBH, SD‑QA, MMSU) and **instruction‑following** (IFEval), which are also strongly tied to the LLM backbone capabilities apart from speech processing quality, suggesting clear headroom for improvement via a stronger LLM.
> > >
> > > We will incorporate this VoiceBench breakdown as an additional baseline row in the main results table for clarity. We are happy to provide further analyses or additional experiments as requested.

---

> ### Author Response · Authors · 2025-11-24
>
> **Part 2/4**
>
> ---
>
> **Weakness 3 & Question 3**: Clarifying the Design of $L _ {\mathrm{align}}$
>
> > Lack of clarity in the design of $L _ {align}$ as a combination of L1 loss and cosine similarity loss (see next section).
>
> > The authors propose to use $L _ \mathrm{align}$ as a combination of L1 loss and cosine similarity loss such that non-textual and textual features can be decoupled. However, the effect or benefit of this is not clear from table 4. Although we see that non-textual features like emotion and gender information is preserved, we do not know if this can be achieved from just a simple L1 loss or cosine-similarity loss.
>
> **Response to Weakness 3 & Question 3**: We thank the reviewer for noting that the design rationale of $L _ {\mathrm{align}}$ was not sufficiently clear in the original text. The motivation for defining $L _ {\mathrm{align}}$ as a **combination of L1 loss and cosine similarity loss** comes from our preliminary experiments, where we found that using only one of these losses failed to achieve efficient audio–text embedding alignment during pretraining.
>
> Following your suggestion, we conducted an **ablation over the loss weights** $\alpha$ (L1) and $\beta$ (cosine similarity). The table below reports OpenAudioBench scores at the end of pretraining:
>
> | Setting                       | AlpacaEval | Llama Questions | TriviaQA | Web Questions |
> | ----------------------------- | ---------- | --------------- | -------- | ------------- |
> | **AlignChat**                 | 80.65      | 81.00           | 60.00    | 60.90         |
> | **No L1 Loss** ($\alpha=0$)   | 10.10      | 0.00            | 0.00     | 0.00          |
> | **No Cosine Loss** ($\beta=0$)| 10.15      | 7.79            | 1.50     | 0.10          |
>
> Removing **either** loss leads to **drastic performance degradation**: most metrics drop near zero (AlpacaEval uses large‑model scoring, with a minimum score of 10). Manual inspection further revealed distinct failure modes:
>
> - **No L1 loss ($\alpha=0$)**: The model could still produce coherent text but failed to understand the input audio embeddings, resulting in irrelevant answers.
>   > **Example Output**: 主任题似乎被截断了，看起来你在提问或者讨论某个主题，但信息不完整。你能提供更多的上下文或者具体问题吗？...
>   > **Translation**: The director's question seems to have been interrupted; it appears you are asking or discussing a topic, but the information is incomplete. Could you provide more context or a specific question? ...
>
> - **No cosine loss ($\beta=0$)**: The outputs were entirely nonsensical, with random or repeated tokens.
>   > **Example Output**: Pleople,P\nPP.Pleople.\nPopleopleople\nPople\n\n\nPopleople\n\nPopleopleopleopleopleople...
>
> These observations confirm that **both losses are essential but play different functional roles**. The cosine similarity term encourages **semantic alignment** even when audio–text embeddings are not perfectly matched in Euclidean space, enabling the LLM to directly consume audio embeddings as input. The intent was **not** to decouple textual and non‑textual features, but rather to ensure robust alignment at the semantic level.

---

> ### Author Response · Authors · 2025-11-24
>
> **Part 3/4**
>
> ---
>
> **Question 1a & Question 1c & Question 1d**: Potential Misalignment during Inference
>
> > How do the authors deal with the mismatch during training and inference? During training, ground-truth transcripts are available, therefore, there is indeed a one-to-one correspondence between the acoustic and text embeddings. However, there will always be errors during inference, in particular, insertion and deletion errors which prohibit this alignment. How is this taken into account.
>
> > How is $L _ {speech} = L _ {text}$ (line 139) ensured during test time?
>
> > The authors mention in line 264, "During inference, speech inputs are first encoded into audio features and then autoregressively converted into speech representations that maintain a strict correspondence with text tokens". That cannot be true in the presence of ASR errors. Please re-phrase.
>
> **Response to Question 1a & Question 1c & Question 1d**: We thank the reviewer for pointing out this important issue. Our original phrasing was indeed insufficiently precise. We will revise the text to state that, **during inference, the correspondence between audio and text tokens is *approximate one‑to‑one* rather than perfectly strict**.
>
> In practice, inference is performed by the speech tokenizer, which autoregressively generates token sequences. Owing to the next‑token prediction training objective, this process tends to preserve close alignment between speech and text token embeddings. However, we acknowledge that insertion or deletion errors during inference can break exact correspondence, particularly in the absence of ground‑truth transcripts.
>
> *Empirical check on alignment drift.*
>
> To quantify this, we computed the **normalized token edit distance** between the inferred token sequence (from greedy decoding) and the ground‑truth transcript, on both training and evaluation splits of OpenAudioBench:
>
> | Dataset             | Normalized Edit Distance (%) |
> | ------------------- | ---------------------------- |
> | Training Dataset    | 6.34                         |
> | Evaluation Dataset  | 3.73                         |
>
> These values indicate that differences are relatively small for the majority of inputs, supporting our claim of **near one‑to‑one alignment** in practical use.
>
> At the same time, the non‑zero edit distance confirms that there is room for improvement. We believe that exploring more advanced alignment algorithms, *e.g.*, incorporating normalization strategies, contrastive objectives, or consistency regularization, could further enhance AlignChat's robustness to inference‑time errors, and we plan to investigate these directions in future work.
>
> ---
>
> **Question 1b**: Performance Gain with Beam Search
>
> > The authors suggest that they use greedy decoding during inference from the AED model. How does this compare with beam search decoding apart from the obvious latency differences?
>
> **Response to Question 1b**: We thank the reviewer for pointing out a potential avenue for improvement. We evaluated the impact of replacing **greedy decoding** with **beam search decoding** (we use beam_size=4) in the speech tokenizer during inference, on OpenAudioBench:
>
> | Decoding Method | AlpacaEval | Llama Questions | TriviaQA | Web Questions |
> | --------------- | ---------- | --------------- | -------- | ------------- |
> | Greedy decoding | 81.26      | 80.33           | 61.40    | 65.90         |
> | Beam search     | 81.21      | 80.33           | 61.20    | 66.20         |
>
> As shown, beam search produces only marginal and inconsistent changes across subsets. We believe this is because the speech tokenizer is initialized from a **pretrained Whisper model** and, after fine‑tuning, already achieves very high accuracy in generating the audio‑aligned text tokens. In such a scenario, the higher‑cost beam search has limited headroom to improve output quality. That is to say, we found that the final speech token sequences during inference is mostly the same with or without beam search.
>
> To further investigate, we manually analyzed the 59 errors made by the model on the **Llama Questions** subset. The breakdown is as follows:
> - **49** errors: the LLM could not answer correctly even with the correct text (LLM‑intrinsic limitation)
> - **2** errors: incorrect token sequence generation by the speech tokenizer
> - **8** errors: misalignment between audio and text embeddings
>
> This confirms that most remaining errors stem from **LLM limitations** or **cross‑modal embedding alignment quality**, rather than from the speech tokenizer's decoding strategy. Therefore, we believe that using a **stronger LLM backbone** and improving **audio–text embedding alignment** would have a far greater impact on performance than changing from greedy to beam search in the audio model.

---

> ### Author Response · Authors · 2025-11-24
>
> **Part 4/4**
>
> ---
>
> **Question 2**: Reason for Using the Modality Projector
>
> > In line 156, "The modality projector decouples transcription from representation alignment to avoid cross-task interference". Does the speech decoder not get adapted with the alignment? If so, how is it decoupled?
>
> **Response to Question 2**: We thank the reviewer for pointing out this area of unclear wording. The motivation for introducing an additional **Modality Projector** is based on our preliminary findings: using the *speech tokenizer's raw output embeddings* for both (i) alignment with the LLM's input space and (ii) speech transcription led to **inferior performance**.
>
> To avoid **cross‑task interference** between transcription and representation alignment, we designed the pipeline so that the speech tokenizer's output is first passed through a **modality projector**. The projected embeddings are then aligned with the LLM's text‑token embeddings, while transcription remains a separate objective for the speech tokenizer itself.
>
> We provide the evaluation results on OpenAudioBench with and without the modality projector:
>
> | Setting                  | AlpacaEval | Llama Questions | TriviaQA | Web Questions |
> | ------------------------ | ---------- | --------------- | -------- | ------------- |
> | **AlignChat**            | 81.26      | 80.33            | 61.40    | 65.90         |
> | **Without Modality Projector** | 81.86      | 77.00            | 61.10    | 61.30         |
>
> The results show that including the modality projector improves **overall performance on the evaluation benchmark**, particularly on subsets where alignment quality is critical. This supports our design choice to **decouple transcription from alignment** via an intermediate projection step.
>
> ---
>
> **Question 4**: Way to Use Dialogue Data
>
> > From table 1, it seems that the authors use dialogue data for stage 1 training. How is a non-ASR data used for stage 1, since it only involves the AED ASR model and the modality adapter?
>
> **Response to Question 4**: We thank the reviewer for pointing out this lack of clarity in our description. The dialogue datasets we use contain both the **spoken audio** and the **corresponding textual transcripts** for each conversation turn. For Stage 1 pretraining, we simply *split each turn of the dialogue into an independent speech–text pair*, which can then be used directly as ASR‑style training data for the AED speech tokenizer and modality adapter.
>
> ---
>
> **Question 5**: Way to use Self-Generated Responses
>
> > In line 231, "for each speech input, the paired text is fed into the LLM backbone to generate the response representing its original behavior." At what stage is this data used?
>
> **Response to Question 5**: We thank the reviewer for pointing out this lack of clarity. The **LLM‑generated responses** are used during **Stage 2** of training. In particular, a distinguishing feature of AlignChat is that its end‑to‑end Stage 2 training employs **responses generated by the LLM itself**, rather than using the responses in the original datasets. This allows the model to directly optimize the *alignment between audio and text input embeddings* in an end‑to‑end manner, while preserving the original behavior of the pretrained LLM backbone.
>
> We provide an ablation experiment showing that Stage 2 finetuning self-generated LLM responses **yields significantly better results**.
>
> | Stage 2 Trainnig Data                  | AlpacaEval | Llama Questions | TriviaQA | Web Questions |
> | ------------------------ | ---------- | --------------- | -------- | ------------- |
> | **Self-generated LLM responses**            | 81.26      | 80.33            | 61.40    | 65.90         |
> | **Original responses in datasets** | 55.68      | 81.00            | 59.70    | 61.50         |
>
> ---
>
> > **Reference**:
> >
> > [1] URO-Bench: Towards Comprehensive Evaluation for End-to-End Spoken Dialogue Models.

---

> > ### Comment · Reviewer_NsNf · 2025-11-25
> >
> > Unfortunately, the authors mis-understood question 2. I wasn't asking for an ablation study for the projector. I understand its role.  My question is how does the "modality projector decouples transcription from representation alignment"? From figure 1, it seems like the decoder will be adapted with the alignment loss + asr loss. Then, how is it decoupling the two tasks. That line would make sense if the modality projector is trained using the alignment loss $\textbf{only}$.

---

> > > ### Author Response · Authors · 2025-11-26
> > >
> > > **Question 1**: Rephrase for clarification
> > >
> > > > ... My question is how does the "modality projector decouples transcription from representation alignment"? From figure 1, it seems like the decoder will be adapted with the alignment loss + asr loss. Then, how is it decoupling the two tasks. That line would make sense if the modality projector is trained using the alignment loss *only*.
> > >
> > > **Response to Question 1**: We thank the reviewer for this follow‑up question and the opportunity to make our methodology clearer.
> > >
> > > In our implementation, the **modality projector** is trained **only** with the alignment loss, while the speech representations *before* the modality projector are used to compute the ASR loss. However, the alignment loss is **not** restricted to the modality projector alone — the speech decoder is indeed updated by both losses (alignment loss through the modality projector; we do not detach the gradient). This means our earlier phrasing in the paper was not sufficiently precise, and we will revise it to state:
> > >
> > > > “We introduce an additional modality projector to alleviate conflict between tasks, as preliminary experiments showed this structure yields better results.”
> > >
> > > The key point is that the projector acts as a *lightweight transformation layer* to bridge the acoustic space optimized for transcription with the semantic space of the LLM, while allowing *joint adaptation* of the decoder parameters by both ASR and alignment objectives. This separation in the transformation stage reduces harmful gradient interference observed in our preliminary setups without the projector.
> > >
> > > We will update the manuscript to explicitly describe this training flow. We are happy to discuss and clarify this aspect further if the reviewer has additional questions.

---

> ### Comment · Reviewer_NsNf · 2025-11-25
>
> The ablations on L2/cosine loss really puzzle me. Removing either L1 or cosine makes the model break down completely. I would not be surprised if some tasks get better performance and others get bad performance on removing L1 or cosine, but the fact that all tasks get so drastically affected, hints to me that there is something else going on.
> ```
> These observations confirm that both losses are essential but play different functional roles.
> ```
> From the ablations, it seems that these losses don't play any role on their own.
>
> This experiment raised further questions for me than answers. I would urge the authors to dig deeper as to what is happening and also correct me if I'm missing anything.
>
> Furthermore, it would be good to add discussions on this phenomenon in the paper.

---

> > ### Author Response · Authors · 2025-11-26
> >
> > **Qusetion 2**: Effect of Removing L1 or Cosine Similarity Loss
> >
> > > The ablations on L2/cosine loss really puzzle me. Removing either L1 or cosine makes the model break down completely...
> > >
> > > This experiment raised further questions for me than answers. I would urge the authors to dig deeper as to what is happening and also correct me if I'm missing anything.
> > >
> > > Furthermore, it would be good to add discussions on this phenomenon in the paper.
> >
> > **Response to Question 2**: We thank the reviewer for raising this important question, which indeed reveals an intriguing phenomenon.
> >
> > The observed **complete breakdown** when removing either L1 or cosine similarity in our ablation is because the reported numbers are from models that undergo **Stage 1 Pretraining only**. AlignChat’s pretraining aligns audio and text embeddings using *only the LLM’s input/output embedding layers*, without supervision from an end‑to‑end language modeling loss. In Stage 1, we design the two losses to play complementary roles:
> >
> > - **Cosine similarity loss** aligns vector *direction*, but, due to normalization in its computation, *does not constrain the embedding norm*. With cosine only, outputs may be syntactically coherent text but semantically unrelated to the speech content. In fact, Stage 1 audio–text embedding cosine similarity averaged only **0.1** on OpenAudioBench.
> > - **L1 loss** aligns both *norm* and *direction*, but alone tends to collapse into meaningless repetitive tokens, with average cosine similarity only **0.2** in Stage 1.
> >
> > From embedding norm statistics, we observe that:
> > 1. **Cosine only** - embedding norms grow rapidly, hindering optimization.
> > 2. **L1 only** - norm and direction constrained but fails to induce robust semantic alignment.
> >
> > We suspect two contributing factors:
> > 1. In the ablation, we reused the original training hyperparameters without re‑tuning for the altered loss function.
> > 2. As discussed in Question 1, alignment and ASR losses are not fully decoupled so that ASR losses may hinder the alignment.
> >
> > For completeness, here are results after **both Stage 1 and Stage 2**:
> >
> > *After Stage 1 Pretraining*
> > | $\alpha : \beta$ | AlpacaEval | Llama Questions | TriviaQA | Web Questions |
> > |------------------|------------|-----------------|----------|---------------|
> > | $1.0 : 5.0$        | 80.65      | 81.00           | 60.00    | 60.90         |
> > | $0.0 : 5.0$        | 10.10      | 0.00            | 0.00     | 0.00          |
> > | $1.0 : 0.0$        | 10.15      | 7.79            | 1.50     | 0.10          |
> >
> > *After Stage 2 Intruction Finetuning*
> > | $\alpha : \beta$ | AlpacaEval | Llama Questions | TriviaQA | Web Questions |
> > |------------------|------------|-----------------|----------|---------------|
> > | $1.0 : 5.0$        | 81.26      | 80.33           | 61.40    | 65.90         |
> > | $0.0 : 5.0$        | 82.11      | 74.67           | 61.20    | 62.90         |
> > | $1.0 : 0.0$        | 56.73      | 54.33           | 17.90    | 40.40         |
> >
> > As with Stage 1, Stage 2 shows that removing either loss damages performance, particularly removing L1, which aligns with our hypothesis that **both losses are essential to achieve robust embedding alignment in our current design**. We agree this phenomenon warrants deeper investigation and will add a dedicated discussion in the revision.
> >
> > We are happy to explore this behaviour further and address any additional questions the reviewer might have.

---

### Official Review · Reviewer_zX26 · 2025-11-01

**Soundness:** 3
**Presentation:** 3
**Contribution:** 2
**Rating:** 4
**Confidence:** 3

**Summary:**

This paper introduces AlignChat, a framework to add end-to-end speech understanding capabilities to large language models (LLMs) without finetuning the LLM backbone.
The key idea is to solve the frequency mismatch between audio and text tokens to enable cross-modal knowledge transfer without requiring finetuning of the LLM. Here, they use an encoder-decoder speech tokenizer based on Whisper features; this architecture autoregressively generates a sequence of speech representations that has a strict one-to-one token-level correspondence with the LLM's text tokens.
This alignment, refined with a two-stage training process, allows the model to achieve state-of-the-art speech-to-text chat performance using a fraction of the training data of compared methods (about 1/20th).

**Strengths:**

The results are pretty striking: AlignChat achieves improved performance on OpenAudioBench and VoiceBench using an order of magnitude less speech data than other methods compared (~15K hours). Aligning the speech tokenizer to the LLM representation space without a forward pass of the LLM also reduces computational overhead compared to prior approaches. Components are ablated and also broken out by different tasks within benchmarks to show variance and generalization vs limitations

**Weaknesses:**

- AlignChat employs an encoder–decoder speech tokenizer initialized with Whisper, but only cites the hours of training data used for AlignChat training for comparison, while most of the compared models are trained from scratch - this should likely be clarified in text for a fair description
- The model critiques cascaded ASR->LLM pipelines for their latency but introduces its own complex, two-step autoregressive process. Inference now requires running a full autoregressive speech decoder to get the representations, and then running the (frozen) LLM to get the final answer. This could also introduce significant latency, but no inference speed or latency benchmarks are provided.

**Questions:**

- How much heavy lifting is the pre-trained Whisper encoder doing? I'd love to see a comparison that replaces the frozen Whisper encoder with a non-text-supervised encoder (such as HuBERT) to show how much of the perfromance gains are from the proposed metrci true sources of the performance gains

- What is the real-world inference latency? A key motivation for end-to-end models is speed. It would be crucial to see a comparison of AlignChat's end-to-end latency (time from speech input to final text token) versus a standard ASR->LLM pipeline and an encoder-only model.

- The probing experiments show that non-textual features like emotion and gender are preserved in the speech representations. Could these preserved features be actively used? For example, could the model be prompted to describe the speaker's emotion, or could the LLM's response style be conditioned on the detected emotion, as hinted at in the appendix ?

---

> ### Author Response · Authors · 2025-11-24
>
> **Part 1/2**
>
> ---
>
> We sincerely thank the reviewer for taking the time to review our paper. We appreciate that you found our approach achieves improved performance on OpenAudioBench and VoiceBench using an order of magnitude less speech data than other methods.
>
> ---
>
> **Weakness 1**: Clarify the Training Hours
>
> > AlignChat employs an encoder–decoder speech tokenizer initialized with Whisper, but only cites the hours of training data used for AlignChat training for comparison, while most of the compared models are trained from scratch - this should likely be clarified in text for a fair description.
>
> **Response to Weakness 1**: We thank the reviewer for highlighting this important point. We agree that clearer wording in the main text will help ensure fair comparison, and we will explicitly clarify this in the revised manuscript.
>
> We also respectfully note that our strongest baseline, *Kimi‑Audio*, adopts a similar initialization strategy: it uses a *pretrained Qwen2.5‑7B* model to initialize the LLM component and a *pretrained Whisper* model for the audio encoder. The key advantage of AlignChat is that we additionally initialize the **Whisper decoder** as part of our speech tokenizer. This preserves more of the pretrained Whisper model's capabilities end‑to‑end, thereby enabling more effective transfer into our encoder–decoder speech tokenizer architecture and contributing to the strong data‑efficient performance we observe.
>
> ---
>
> **Weakness 1 & Question 2**: Inference Latency Benchmarks
>
> > The model critiques cascaded ASR->LLM pipelines for their latency but introduces its own complex, two-step autoregressive process. Inference now requires running a full autoregressive speech decoder to get the representations, and then running the (frozen) LLM to get the final answer. This could also introduce significant latency, but no inference speed or latency benchmarks are provided.
>
> > What is the real-world inference latency? A key motivation for end-to-end models is speed. It would be crucial to see a comparison of AlignChat's end-to-end latency (time from speech input to final text token) versus a standard ASR->LLM pipeline and an encoder-only model.
>
> **Response to Weakness 1 & Question 2**: We thank the reviewer for raising this important point, **real‑world inference latency** is indeed a crucial metric for evaluating speech‑to‑text dialogue systems. To quantify this, we benchmarked AlignChat against a cascaded ASR + LLM pipeline on a single NVIDIA A800 GPU.
>
> | Model         | First‑Token Latency (ms)     | Parameters (M) | GPU Memory (GB) |
> | ------------- | ---------------------------- | -------------- | --------------- |
> | ASR + LLM      | $367.4 \pm 20.4$           | 771            | 19.45           |
> | AlignChat     | $270.5 \pm 15.2$           | 731            | 19.22           |
>
> AlignChat exhibits a **substantially lower first‑token latency** compared to the cascaded baseline. We attribute this to fundamental architectural differences: in common ASR systems (e.g., Whisper), the ASR tokenizer and LLM tokenizer differ, requiring the ASR model to output a **full text transcription**, which is then re‑tokenized into LLM input tokens. This sequential process prevents efficient parallelization between the audio and text modules, and demands additional engineering to optimize latency‑sensitive metrics like first‑token delay.
>
> In contrast, AlignChat **re‑uses the LLM's tokenizer for the speech**, enabling it to produce speech embeddings that are **directly consumable by the LLM**. This allows the speech tokenizer and LLM to operate in parallel during inference, reducing startup latency without compromising alignment quality or response accuracy.

---

> ### Author Response · Authors · 2025-11-24
>
> **Part 2/2**
>
> ---
>
> **Question 1**: Identifying the Source of the Performance Gains
>
> > How much heavy lifting is the pre-trained Whisper encoder doing? I'd love to see a comparison that replaces the frozen Whisper encoder with a non-text-supervised encoder (such as HuBERT) to show how much of the perfromance gains are from the proposed metrci true sources of the performance gains
>
> **Response to Question 1**: We thank the reviewer for this thoughtful question. We believe that much of AlignChat's performance advantage over other baselines stems from our effective transfer of capabilities from both the **pretrained Whisper** speech model and the **pretrained LLM**, especially given that AlignChat is trained with **an order of magnitude less speech data** compared to many baselines.
>
> Replacing the pretrained Whisper encoder with a non‑text‑supervised encoder (e.g., HuBERT) would require training the corresponding decoder **from scratch**. Based on our previous attempts to train a decoder from scratch on Whisper's encoder, this leads to a **substantial performance drop**.
>
> *OpenAudioBench Results:*
>
> | Setting                           | AlpacaEval | Llama Questions | TriviaQA | Web Questions |
> | --------------------------------- | ---------- | --------------- | -------- | ------------- |
> | Using Pretrained Whisper Decoder  | 80.65      | 81.00            | 60.00    | 60.90         |
> | Training Decoder from Scratch     | 47.34      | 45.00            | 10.10    | 30.80         |
>
> As shown, the absence of the pretrained decoder severely degrades performance across all subsets.
>
> That said, we agree that in settings with **greater training resources**, using a non‑text‑supervised encoder could potentially preserve **more raw acoustic information** than a text‑supervised one, making this a promising direction for future exploration. This would allow us to examine the trade‑off between **retaining richer audio features** and maintaining strong **speech‑to‑text conversational accuracy**.
>
> ---
>
> **Question 3**: LLM's Usage of Emotion Infomation
>
> > The probing experiments show that non-textual features like emotion and gender are preserved in the speech representations. Could these preserved features be actively used? For example, could the model be prompted to describe the speaker's emotion, or could the LLM's response style be conditioned on the detected emotion, as hinted at in the appendix?
>
> **Response to Question 3**: We thank the reviewer for directing our attention to this important concern. To support our claim that *non‑textual audio features such as emotion are preserved and can be actively used* in AlignChat's speech representations, we evaluated the model on the **UnderEmotion‑en** subset of the URO‑Bench dataset [1]. This subset contains speech inputs with diverse emotional tones, and we use an LLM‑based automatic evaluation to judge whether the dialogue model's responses correctly interpret the emotion and respond **empathetically**.
>
> | Model         | UnderEmotion‑en |
> | ------------- | --------------- |
> | ASR + LLM      | 40.31           |
> | AlignChat     | 46.47           |
> | OpenS2S v1 [2] | 46.90           |
>
> Here, the **ASR + LLM** baseline converts speech entirely to text before LLM processing, thereby discarding emotional cues entirely. **OpenS2S v1** [2] is an encoder‑only, open‑source audio‑dialogue model specifically designed for empathetic responses.
>
> It is encouraging that **AlignChat achieves performance very close to OpenS2S v1** on this emotion‑specific benchmarkm despite using fewer audio tokens and being optimized for general speech‑to‑text chat. This result supports two key points:
> 1. The performance bottleneck from reducing LLM input token count in speech tasks is **not significant** for emotion understanding.
> 2. AlignChat's unified architecture has promising potential to extend to **tasks where emotion detection and conditioning actively inform the response style**.
>
> ---
>
> > **Reference**:
> >
> > [1] URO-Bench: Towards Comprehensive Evaluation for End-to-End Spoken Dialogue Models.
> >
> > [2] OpenS2S: Advancing Fully Open-Source End-to-End Empathetic Large Speech Language Model.

---

### Official Review · Reviewer_8Srw · 2025-11-03

**Soundness:** 2
**Presentation:** 3
**Contribution:** 2
**Rating:** 2
**Confidence:** 4

**Summary:**

The paper proposes AlignChat, an E2E speech to text chat framework that enforces one-to-one token-level alignment between speech and text, allowing the model to freeze the LLM backbone and still transfer language ability into the speech task.

More concretely, a Whisper-initialized encoder–decoder speech tokenizer produces a ~3Hz sequence whose per-token representations are mapped into the LLM’s input embedding space, with the LLM’s embedding layer and output head shared across modalities. Training uses two stages: (1) ASR + alignment without running the LLM’s transformer; (2) instruction tuning with LLM-generated targets while still freezing the LLM. Reported results claim competitive or SOTA speech-to-text chat performance on OpenAudioBench and VoiceBench.

**Strengths:**

From the point of speech understanding, it's well-motivated. Framing speech to text as a strict per-token alignment problem and then replacing the LLM’s text embeddings in-place with aligned speech representations is interpretable. Sharing the vocabulary/embedding layer further tightens the bridge.

Compare with Flamingo style of approach, it reduced the computation as the input frequency of audio tokens are much lower.

**Weaknesses:**

My main concern is the encoder-decoder tokenizer 1-1 assumption introduced a performance bottleneck to the proposed approach. LLM has no way to fix any mistake made by the enc-dec. For audio like music, such representation is for sure lossy.

The claim for the model still keep audio information are weak. Gender are very easy to be captured by any encoder. Emotion need to compare with proper baseline. There is no quantitative experiments about if speaker information has been preserved besides t-SNE plot.

**Questions:**

Does this approach applicable to multilingual?

Do you compared with a simple baseline, using ASR transcription + an pre computed audio embedding?

Do you have error analysis, when the alignment failed or the aligner halluciate?

---

> ### Author Response · Authors · 2025-11-24
>
> **Part 1/3**
>
> ---
>
> We sincerely thank the reviewer for taking the time to review our paper. We appreciate that you found our approach well-motivated and can reduce the overall computation.
>
> ---
>
> **Weakness 1**: Concern about Performance Bottleneck
>
> > My main concern is the encoder-decoder tokenizer 1-1 assumption introduced a performance bottleneck to the proposed approach. LLM has no way to fix any mistake made by the enc-dec. For audio like music, such representation is for sure lossy.
>
> **Response to Weakness 1**: We thank the reviewer for this insightful observation. We acknowledge that introducing a *one‑to‑one correspondence between audio and text tokens* can reduce the number of LLM input tokens derived from audio, and in principle may act as a performance bottleneck. However, we view this as a deliberate **efficiency–effectiveness trade‑off**. In the specific domain addressed by our work, *speech‑to‑text conversational tasks*, preserving the **semantic content** of speech to generate accurate responses is paramount, while richer acoustic details mainly play a supporting role. We respectfully note that in our target setting, this bottleneck **does not lead to significant degradation** in overall performance. Evaluating AlignChat on more general audio (e.g., music) is indeed an interesting future research direction.
>
> *LLM compensation for tokenizer errors.*
> Interestingly, our manual inspection revealed several cases where the LLM compensated for errors from the speech tokenizer (see Appendix A.2.6). For example:
>
> > **Question**: How do I clean my armpits?
> >
> > **ASR Output**: How do I clean my *AR pits*?
> >
> > **ASR + LLM Response**: *AR (Augmented Reality) "pits"* likely refers to the lenses or windows of your AR glasses or headset. Cleaning these lenses is important...
> >
> > **AlignChat Response**: *Cleaning underarm areas, often referred to as armpits*, is important for maintaining hygiene and preventing odor. Here's a step-by-step guide...
>
> In this case, the cascaded ASR + LLM pipeline propagated a transcription error, while AlignChat, leveraging the pretrained LLM's robust comprehension and reasoning abilities, was able to **infer the correct meaning directly from audio embeddings** and produce the intended answer.
>
> *Comparative analysis.*
> To better position AlignChat relative to both cascaded ASR + LLM and encoder‑only end‑to‑end methods, we report performance, latency, and resource usage on a single NVIDIA A800 GPU:
>
> | Model         | OpenAudioBench Avg. | First‑Token Latency (ms)     | Parameters (M) | GPU Memory (GB) |
> | ------------- | ------------------- | ---------------------------- | -------------- | --------------- |
> | ASR + LLM      | 71.95               | $367.4 \pm 20.4$            | 771            | 19.45           |
> | Encoder‑only  | 71.92               | $292.0 \pm 12.7$            | 625            | 18.74           |
> | AlignChat     | 72.22               | $270.5 \pm 15.2$            | 731            | 19.22           |
>
> We view AlignChat as offering a **balanced trade‑off**:
>
> 1. Compared to cascaded ASR + LLM, AlignChat provides an **end‑to‑end capability** by removing the intermediate transcription stage, allowing the speech and text modules to run in parallel, improving efficiency and first‑token latency.
> 2. Compared to encoder‑only end‑to‑end approaches, AlignChat uses **fewer audio tokens**, requiring lower training cost while achieving superior performance on *speech‑centric tasks*. Notably, as LLM size grows, encoder‑only methods incur higher inference latency and computational overhead due to more audio tokens, whereas AlignChat's pretraining requires only the LLM's embedding layers, making it a more **cost‑efficient choice** for both training and deployment.

---

> ### Author Response · Authors · 2025-11-24
>
> **Part 2/3**
>
> ---
>
> **Weakness 2**: Proof of Preserving Audio Information.
>
> > The claim for the model still keep audio information are weak. Gender are very easy to be captured by any encoder. Emotion need to compare with proper baseline. There is no quantitative experiments about if speaker information has been preserved besides t-SNE plot.
>
> **Response to Weakness 2**: We thank the reviewer for highlighting the need for stronger quantitative evidence to support our claim that audio information, beyond pure text content, is preserved in the audio token representations of AlignChat.
>
> To this end, we evaluated our model on the **UnderEmotion‑en** subset of the URO‑Bench dataset [1]. This subset contains speech inputs with varying emotional tones, and we use an LLM‑based automatic evaluation to judge whether the dialogue model's responses correctly interpret the emotion present in the speech and respond **empathetically**.
>
> | Model         | UnderEmotion‑en |
> | ------------- | --------------- |
> | ASR + LLM      | 40.31           |
> | AlignChat     | 46.47           |
> | OpenS2S v1 [2] | 46.90           |
>
> Here, the **ASR + LLM** baseline converts speech entirely to text before LLM processing, thus discarding any explicit emotional cues. **OpenS2S v1** [2] is an encoder‑only end‑to‑end audio‑dialogue model specifically designed to deliver empathetic responses.
>
> It is encouraging to see that **AlignChat achieves performance very close to OpenS2S v1** on this emotion‑specific benchmark, despite using fewer audio tokens. This result supports two points:
> 1. **Minimal performance bottleneck** — Reducing the number of LLM input tokens from audio does not significantly impair emotion‑related performance in our speech‑centric tasks.
> 2. **Potential for task generalization** — AlignChat's unified speech–LLM pipeline can be extended effectively to tasks beyond conventional speech‑to‑text, such as empathetic dialogue generation.
>
> ---
>
> **Question 1**: Multilingual Applicability
>
> > Does this approach applicable to multilingual?
>
> **Response to Question 1**: We thank the reviewer for this insightful question. Yes, the proposed approach is readily applicable to multilingual settings. The *Qwen tokenizer* used in our framework is already well‑adapted for multilingual text processing. By fine‑tuning AlignChat on **multilingual speech–text datasets**, we can efficiently build speech‑to‑text dialogue models that handle multiple languages within the same unified architecture. This design choice ensures that our method can extend beyond monolingual use cases with minimal modification to the pipeline.
>
> ---
>
> **Question 2**: Comparison with Simple Baselines
>
> > Do you compared with a simple baseline, using ASR transcription + an pre computed audio embedding?
>
> **Response to Question 2**: We thank the reviewer for this constructive suggestion. To address your point, we compared **AlignChat** against a simple baseline that uses **ASR transcription plus a pre‑computed audio embedding**. For this, we selected **DeSTA2** [3], published at ICASSP 2025, which fits the description: it feeds the textual representation from ASR into the LLM and appends an additional audio embedding as part of the LLM input.
>
> | Model     | AlpacaE. | CommonE. | WildVoice | SD‑QA  | MMSU   | OBQA   | BBH    | IFEval | AdvBench | Avg.   |
> | --------- | -------- | -------- | --------  | ------ | ------ | ------ | ------ | ------ | -------- | ------ |
> | DeSTA2    | 4.36     | 3.33     | 3.42      | 56.06  | 66.43  | 61.98  | 66.20  | 52.63  | 98.27    | 69.31  |
> | AlignChat | 4.57     | 4.13     | 4.21      | 62.84  | 69.65  | 85.49  | 69.60  | 63.17  | 99.42    | 78.71  |
>
> As shown above, AlignChat achieves **consistently higher performance across nearly all VoiceBench subsets** compared to DeSTA2. We attribute this improvement to AlignChat's design, which **preserves the original capabilities of the LLM** by avoiding substantial changes to its input format, thereby preventing the degradation in performance that can occur when textual input is significantly altered.

---

> ### Author Response · Authors · 2025-11-24
>
> **Part 3/3**
>
> ---
>
> **Question 3**: Requires for Error Analysis
>
> > Do you have error analysis, when the alignment failed or the aligner halluciate?
>
> **Response to Question 3**: We thank the reviewer for this constructive suggestion. To better understand failure cases where alignment breaks down or the aligner hallucinates, we manually analyzed model outputs on the **Llama Questions** subset of OpenAudioBench, categorizing errors by type.
>
> We identified three main categories:
> 1. **LLM‑intrinsic limitation**: Cases where the LLM itself cannot answer correctly, and given that AlignChat freezes the LLM during training, it inherits this limitation.
> 2. **Token correspondence errors**, Cases where the speech tokenizer's predicted text sequence differs from the ground‑truth transcript, breaking the one‑to‑one correspondence and leading to an incorrect answer.
> 3. **Embedding misalignment**, Cases where the predicted text sequence matches the ground truth, but the audio and text embeddings are insufficiently aligned, causing the LLM to produce an incorrect answer.
>
> Across the 199 questions in this subset, AlignChat produced 59 incorrect answers. Their distribution is as follows:
>
> | LLM Limitation | Token Correspondence Error | Embedding Misalignment |
> | -------------- | ------------------------- | ---------------------- |
> | 49             | 2                         | 8                      |
>
> The majority of errors stem from the LLM's own performance limits. Between categories 2 and 3, embedding misalignment has the greater negative impact on AlignChat compared to explicit token correspondence errors. This highlights **room for further improvement** in our alignment training procedure.
>
> Encouragingly, we found 5 cases where the base LLM failed to answer correctly, yet AlignChat produced the correct response. This suggests that AlignChat possesses a degree of **generalization capability**, and may even elicit latent abilities within the LLM that did not manifest in the baseline setting.
>
> ---
>
> > **Reference**:
> >
> > [1] URO-Bench: Towards Comprehensive Evaluation for End-to-End Spoken Dialogue Models.
> >
> > [2] OpenS2S: Advancing Fully Open-Source End-to-End Empathetic Large Speech Language Model.
> >
> > [3] Developing Instruction-Following Speech Language Model Without Speech Instruction-Tuning Data. ICASSP 2025.

---

### Official Review · Reviewer_XoS6 · 2025-11-04

**Soundness:** 2
**Presentation:** 3
**Contribution:** 2
**Rating:** 2
**Confidence:** 4

**Summary:**

The paper introduces AlignChat, a framework that integrates speech understanding into large language models (LLMs) via token-level representation alignment. Unlike prior end-to-end systems that rely on encoder-only speech tokenizers or require LLM fine-tuning (which risks catastrophic forgetting), AlignChat uses an encoder–decoder speech tokenizer designed to achieve precise one-to-one alignment between speech and text tokens.

Key contributions of the paper include:
1. encoder-decoder speech tokenizer: unlike most of the prior work in speech LLMs, this paper proposes to employ an encoder-decoder speech tokenizer, initialized from whisper-larger-v3.
2. two-stage training strategy: the first step aims to achieve exact token-level alignment between speech and text, and the second stage performs end-to-end instruction tuning.
3. Data synthesis: the paper regenerates the text response from available speech datasets to ensure consistent responses to speech and text input.

**Strengths:**

1. token-level alignment: Introduces an encoder–decoder speech tokenizer that aligns each speech token one-to-one with text embeddings, unlike prior coarse audio-token approaches.
2. No LLM update needed: the LLM remains frozen, avoiding catastrophic forgetting while still enabling end-to-end speech understanding.
3. Data efficiency: Achieves SOTA results on OpenAudioBench and VoiceBench using only ~1/20 data of prior work.

**Weaknesses:**

I would like the authors to explain how this framework is different from cascading ASR model with an LLM (perhaps plus joint optimization using the LLM prediction loss). In section 3.1.1, it's mentioned:
> For reproducibility, we adopt greedy decoding for both the generation of speech representations and the final text outputs.

I am confused what the speech decoder actually generates during inference and the authors didn't give any more details about it. Hence, the model is built upon whisper and Qwen2.5, in Table 2, there's supposed to be a cascaded system baseline using Whisper + Qwen2.5 (or for more fair comparison, finetune the Whisper on the 15k hour speech datasets) in order to validate the efficacy of the proposed finetuning method. In the performance comparison, it'd be also more informative to compare with some of the commercial multimodal LLMs such as Gemini/GPT4o.

Without the differentiators with cascaded system or comparison with cascaded system, I cannot judge whether AlignChat’s “end-to-end” gains are due to its alignment design or simply comparable to a well-tuned cascaded system. Therefore I have to rate the paper at 2. I am open to discuss more with the authors.

**Questions:**

find in 'Weakness' section

---

> ### Author Response · Authors · 2025-11-24
>
> **Part 1/2**
>
> ---
>
> We sincerely thank the reviewer for taking the time to review our paper. We appreciate that you found our method aligns speech and text tokens to achieve high data efficiency.
>
> ---
>
> **Weakness 1**: Explain the Difference from the Cascaded Baseline
>
> > I would like the authors to explain how this framework is different from cascading ASR model with an LLM (perhaps plus joint optimization using the LLM prediction loss) ... I am confused what the speech decoder actually generates during inference and the authors didn't give any more details about it.
>
> **Response to Weakness 1**: We thank the reviewer for raising this thought‑provoking question. The fundamental distinction between *AlignChat* and a conventional cascaded ASR + LLM pipeline lies in the *interface between the speech module and the language model*: in AlignChat, the speech tokenizer passes **intermediate audio embeddings** directly to the LLM, rather than transcribed text. This makes AlignChat an **end‑to‑end framework**, whereas cascaded baselines must first generate text via ASR and then re‑tokenize it for the LLM.
>
> *Inference behavior.*
> During inference, the decoder component of the speech tokenizer produces both (i) audio token embeddings, which are fed into the LLM for direct end‑to‑end question answering, and (ii) a predicted text token sequence, used to maintain approximate one‑to‑one correspondence between audio and text tokens when no ground‑truth text is available at test time.
>
> While cascaded ASR + LLM remains a strong baseline for speech dialogue tasks, existing research has increasingly moved toward end‑to‑end speech‑LLM frameworks. This shift is motivated by two inherent limitations of the cascaded approach:
>
> 1. **Tokenizer mismatch and lack of joint optimization.** Widely used ASR systems (e.g., Whisper) and LLMs employ different tokenizers. As a result, the ASR output must be fully converted to text and then re‑tokenized for the LLM, preventing direct end‑to‑end optimization of the ASR and LLM components, and hindering efficient parallelism — particularly for latency metrics such as first‑token delay.
> 2. **Loss of fine‑grained speech information.** Converting speech to text before LLM processing discards prosody, timbre, emotion, and other subtle acoustic features, limiting the ability to build more **empathetic** and contextually aware dialogue systems.
>
> By contrast, AlignChat uses a **unified tokenizer design** for the audio and LLM components, enabling simpler parallel inference, and **supplies the LLM with rich audio embeddings** rather than ASR‑generated text. This allows the LLM to leverage fine‑grained acoustic information that would be lost in a cascaded pipeline.
>
> To further support these distinctions, we provide in our response to **Weakness 2** a detailed comparison between AlignChat and the cascaded baseline in terms of dataset accuracy, first‑token latency, parameter count, and GPU memory usage.

---

> ### Author Response · Authors · 2025-11-24
>
> **Part 2/2**
>
> ---
>
> **Weakness 2**: Additional Comparison results
>
> > in Table 2, there's supposed to be a cascaded system baseline using Whisper + Qwen2.5 (or for more fair comparison, finetune the Whisper on the 15k hour speech datasets) in order to validate the efficacy of the proposed finetuning method. In the performance comparison, it'd be also more informative to compare with some of the commercial multimodal LLMs such as Gemini/GPT4o.
>
> **Response to Weakness 2**: We sincerely thank the reviewer for this very constructive suggestion. To provide a thorough comparison between a cascaded ASR + LLM system and *AlignChat*, we conducted experiments on a single NVIDIA A800 GPU. The results are summarized below:
>
> | Model              | OpenAudioBench Avg. | UnderEmotion‑en | First‑Token Latency (ms)       | Parameters (M) | GPU Memory (GB) |
> | ------------------ | ------------------- | ---------------- | ------------------------------ | -------------- | --------------- |
> | ASR + LLM           | 71.95               | 40.31            | $367.4 \pm 20.4$              | 771            | 19.45           |
> | AlignChat           | 72.22               | 46.47            | $270.5 \pm 15.2$              | 731            | 19.22           |
>
> Here, **OpenAudioBench Avg.** evaluates overall audio‑to‑text dialogue capabilities. **UnderEmotion‑en** is a subset of the URO‑Bench dataset [1], containing audio inputs with varied emotional tones. We follow the official evaluation code to use an LLM‑based automatic evaluation to assess whether the model can accurately interpret emotion in speech and respond empathetically.
>
> We find that the end‑to‑end AlignChat framework yields **substantial improvements in first‑token latency** and **emotion‑related evaluation** compared to the cascaded ASR + LLM baseline. Interestingly, it also achieves a small gain in general dialogue performance. Beyond possible random noise effects, we observed clear cases where AlignChat performed better due to its direct use of audio embeddings. For example:
>
> > **Question**: How do I clean my armpits?
> >
> > **ASR Output**: How do I clean my *AR pits*?
> >
> > **ASR + LLM Response**: *AR (Augmented Reality) "pits"* likely refers to the lenses or windows of your AR glasses or headset. Cleaning these lenses is important...
> >
> > **AlignChat Response**: *Cleaning underarm areas, often referred to as armpits*, is important for maintaining hygiene and preventing odor. Here's a step-by-step guide...
>
> In this case, the cascaded approach suffered from an ASR transcription error (“AR pits” vs “armpits”), which propagated to the LLM and led to a wrong answer. By contrast, AlignChat's LLM could **infer the correct semantic meaning directly from audio embeddings**, producing the intended response without explicit textual correction.
>
> Finally, per your suggestion, we also provide a comparison with commercial multimodal LLMs (GPT‑4o variants) on VoiceBench:
>
> | Model                   | AlpacaE. | CommonE. | WildVoice | SD‑QA  | MMSU   | OBQA   | BBH    | IFEval | AdvBench | Avg.   |
> | ----------------------- | -------- | -------- | --------  | ------ | ------ | ------ | ------ | ------ | -------- | ------ |
> | GPT‑4o‑Audio            | 4.78     | 4.49     | 4.58      | 75.50  | 80.25  | 89.23  | 84.10  | 76.02  | 98.65    | 86.75  |
> | GPT‑4o‑mini‑Audio       | 4.75     | 4.24     | 4.40      | 67.36  | 72.90  | 84.84  | 81.50  | 72.90  | 98.27    | 82.84  |
> | AlignChat               | 4.57     | 4.13     | 4.21      | 62.84  | 69.65  | 85.49  | 69.60  | 63.17  | 99.42    | 78.71  |
>
> Although AlignChat does not yet match the performance of proprietary commercial speech‑dialogue models, it achieves competitive results considering **its modest 7B parameter size and use of only ~10K hours of speech data for training**. Besides, the largest gaps appear on knowledge-intensive tasks (BBH, SD-QA, MMSU) and instruction-following (IFEval), which are strongly tied to the underlying LLM's capabilities rather than speech processing quality. We believe there is considerable room for further improvement, and these comparisons validate the effectiveness of our finetuning methodology.
>
> ---
>
> > **Reference**:
> >
> > [1] URO-Bench: Towards Comprehensive Evaluation for End-to-End Spoken Dialogue Models.

---

### Official Review · Reviewer_PPnd · 2025-11-07

**Soundness:** 3
**Presentation:** 4
**Contribution:** 2
**Rating:** 6
**Confidence:** 4

**Summary:**

This is essentially an end-to-end speech-to-text chat framework that adds speech understanding to a frozen LLM without hurting its core skills. The main essence is token-by-token alignment - a Whisper-based encoder–decoder generates tokens at about the same pace as text, shares the LLM’s tokenizer and embeddings and output head, with a projector layer to map speech representation to the LLM token input space.

Training methodology

1. ASR style pretraining to align and transcribe without even running the LLM
2. instruction tuning where the LLM itself supplies the targets so that its original text behavior carries over to speech.

The result/contribution is that it's competitive on OpenAudioBench and VoiceBench, shows tight speech–text alignment and keeps useful non-textual cues (like emotion) while avoiding possible catastrophic forgetting.

**Strengths:**

The paper combines known methodologies frozen LLMs, Whisper-based speech encoders, and projector-based alignment methods but adds an encoder–decoder speech tokenizer that matches text token rate and shares the LLM's tokenizer/embeddings/output head. I think that this strict alignment is a good step away from the encoder-only audio-token pipelines. The architecture and training flow are also easy and intuitive to follow + the figures are illustrated well. I like to believe that in general this could become a default recipe for adding speech to LLMs without retraining the backbone, and the rate-matching idea may generalize to other modalities in the future..

**Weaknesses:**

Alignment loss design is heuristic - Report embedding norms and cosine histograms pre/post training; specify whether vectors are normalized before L1/cosine, Ablate α/β, compare to normalized-only (cosine) and contrastive losses (such as for eg InfoNCE with in-batch negatives). token synchronization is the essence of the paper and needs a better metric to justify it. Consider adding something like a token edit distance between inferred token sequence and transcript

**Questions:**

- Please report full benchmark scores for Stage 1 alone, and the incremental gains of Stage 2 per subset.
- Qwen's tokenizer may segment differently from Whisper's. Did you observe systematic failure modes. Any language/domain-specific mitigations?
- How do you ensure Lspeech ≈ Ltext and maintain 1-to-1 correspondence without teacher forcing? Did you use scheduled sampling/professor forcing or any consistency regularizers? Please quantify alignment drift. How robust is the alignment under fillers, repetitions or partial words?
- Can you report an explicit "token synchronization error" metric to substantiate the "strict alignment" claim?

---

> ### Author Response · Authors · 2025-11-24
>
> **Part 1/3**
>
> ---
>
> We sincerely thank the reviewer for taking the time to review. We appreciate that you found our method a good step away from the encoder-only audio-token pipelines, and our method is easy and intuitive to follow. Based on your valuable comments, we will provide detailed feedback.
>
> ---
>
> **Weakness 1**: More Ablation on Loss Design
>
> > Alignment loss design is heuristic - Report embedding norms and cosine histograms pre/post training; specify whether vectors are normalized before L1/cosine, Ablate $\alpha$/$\beta$, compare to normalized-only (cosine) and contrastive losses (such as for eg InfoNCE with in-batch negatives).
>
> **Response to Weakness 1**: We thank the reviewer for this insightful comment and for encouraging us to provide a deeper analysis of our alignment loss design. Your suggestions have helped us strengthen its empirical grounding.
>
> *Implementation Details.*
> Following your request, we clarify that **we do not normalize the vectors** before computing the L1 loss, as our design aims to directly align speech and text token embeddings without altering their absolute magnitudes. In contrast, the cosine similarity computation inherently involves normalization. This choice facilitates a direct one-to-one alignment between speech and text tokens, though we acknowledge that normalization-based alignment could be an interesting avenue for future work.
>
> We have added in the revised appendix (Figure 5, Appendix) the curves for the three losses used during pretraining: $\mathcal{L} _ \mathrm{asr}$, $\mathcal{L} _ \mathrm{L1}$, and $\mathcal{L} _ \mathrm{cos\_sim}^2$. The average loss values for the first and last 300 training steps are shown below:
>
>
> | Training Steps  | $\mathcal{L} _ \mathrm{asr}$ | $\mathcal{L} _ \mathrm{L1}$ | $\mathcal{L} _ \mathrm{cos\_sim}^2$ |
> | --------        | --------  | -------- | --------  |
> | First 300 Steps | 6.256   | 0.2465 | 0.5271  |
> | Last 300 Steps  | 0.2678  | 0.0059 | 0.0652  |
>
> All three losses decrease substantially during training. In particular, the final ASR loss reaches a reasonably low value, $0.2678$, indicating a strong alignment between speech tokens and text tokens after training.
>
> *Embedding Norms and Cosine Histograms.*
> In line with your suggestion, we report the embedding norms and cosine similarity histograms at the end of training for the OpenAudioBench dataset (using 256 randomly-selected samples). Key statistics are summarized below (full visualizations are provided in the revised appendix (Figure 6, Appendix)):
>
> |  | Mean | Median | Std |
> | -------- | -------- | -------- | -------- |
> | Cosine Similarity ($\uparrow$) | 0.8219 | 0.8991 | 0.1748 |
> | L1 Distance ($\downarrow$)  | 31.58  | 23.49  | 21.32 |
>
> A cosine similarity of around 0.82 for high-dimensional embeddings indicates that the speech tokenizer's outputs and the LLM's input representations are very well aligned directionally. The L1 distance is relatively large considering the average L1 norm of a token embedding is $\approx 38.07$, suggesting that $\mathcal{L} _ \mathrm{cos\_sim}^2$ plays a more crucial role in model performance, while $\mathcal{L} _ \mathrm{L1}$ primarily constrains the overall norm of embeddings.
>
> *Ablation on $\alpha$ and $\beta$.*
> We further ablated the loss coefficients $\alpha$ (for L1) and $\beta$ (for cosine similarity). The table below shows performance at the end of pretraining on OpenAudioBench benchmarks:
>
> | $\alpha : \beta$ | AlpacaEval | Llama Questions | TriviaQA | Web Questions |
> | -------- | -------- | -------- | -------- | -------- |
> | $1.0 : 5.0$  | 80.65 | 81.00 | 60.00 | 60.90 |
> | $0.0 : 5.0$  | 10.10 | 0.00 | 0.00 | 0.00 |
> | $1.0 : 0.0$  | 10.15 | 7.79 | 1.50 | 0.10 |
>
> Removing either L1 or cosine loss drastically degrades performance (often near zero), confirming their complementary importance. Manual inspection reveals interesting qualitative differences (please refer to Appendix B.2.1 in the newer revision for examples): with $\alpha = 0.0$ (no L1), the model still generates coherent text but fails to interpret audio representations meaningfully; with $\beta = 0.0$ (no cosine loss), outputs become entirely nonsensical (e.g., repeated or malformed tokens). This suggests **a nuanced role for each loss component in maintaining semantic and structural alignment**.
>
> Due to rebuttal time constraints, we have not yet explored normalization before L1/cosine computation or alternative loss formulations such as InfoNCE with in-batch negatives. We agree these are promising modifications and plan to investigate more sophisticated speech-text alignment approaches in future work.

---

> ### Author Response · Authors · 2025-11-24
>
> **Part 2/3**
>
> ---
>
> **Weakness 2 & Question 4**: Verify Token Synchronization
>
> > Token synchronization is the essence of the paper and needs a better metric to justify it. Consider adding something like a token edit distance between inferred token sequence and transcript.
>
> > Can you report an explicit "token synchronization error" metric to substantiate the "strict alignment" claim?
>
> **Response to Weakness 2 & Question 4**: We thank the reviewer for highlighting the importance of explicitly quantifying token synchronization. This is indeed a central aspect of our approach. Following your valuable suggestion, we have introduced the **normalized token edit distance** between the inferred token sequence (produced via greedy decoding) and the corresponding transcript as a more concrete metric.
>
> The results for both the training set and the evaluation set (OpenAudioBench, using 2048 randomly-selected samples) are summarized below, with full details and distributions provided in the revised appendix (Figure 7, Appendix):
>
> |  | Training Dataset | Evaluation Dataset |
> | -------- | -------- | -------- |
> | Average Normalized Edit Distance (%) | 6.34 | 3.73 |
>
> These results indicate that, while the inferred and reference token sequences are not perfectly identical, the differences are relatively small. This supports our claim that AlignChat achieves **near one-to-one alignment** between audio and text tokens for most inputs.
>
> Interestingly, the normalized edit distance on the evaluation set is even lower than on the training set. This suggests two possible explanations: (i) the continued training from the pretrained base model has led to good generalization; and/or (ii) the evaluation audio data is slightly cleaner or less challenging than the training data.
>
> The fact that the edit distance is nonzero also underscores that there is still room for improvement. We believe that exploring more advanced speech–text alignment algorithms could further reduce the synchronization error, and we consider this a promising direction for our future work.
>
> ---
>
> **Question 1**: Full Benchmark Score for Stage 1
>
> > Please report full benchmark scores for Stage 1 alone, and the incremental gains of Stage 2 per subset.
>
> **Response to Question 1**: We thank the reviewer for this constructive suggestion. We now report the full benchmark results after Stage 1 pretraining, along with the **incremental gains from Stage 2 fine-tuning** for each dataset subset. The tables below summarize the core results.
>
> *OpenAudioBench Results:*
>
> | Training Stage  | AlpacaEval | Llama Questions | TriviaQA | Web Questions |
> | --------        | --------   | --------        | -------- | --------      |
> | After Stage 1   | 80.65      | 81.00           | 60.00    | 60.90         |
> | After Stage 2   | 81.26      | 80.33           | 61.40    | 65.90         |
> | Incremental Gains | +0.61   | −0.67           | +1.40    | +5.00         |
>
> *VoiceBench Results:*
>
> | Training Stage  | AlpacaE. | CommonE. | WildVoice | SD-QA | MMSU  | OBQA  | BBH   | IFEval | AdvBench | Avg.  |
> | --------        | -------- | -------- | --------  | -------- | -------- | -------- | -------- | -------- | -------- | -------- |
> | After Stage 1   | 4.45     | 4.04     | 4.03      | 55.70   | 65.91  | 85.93  | 65.00  | 58.80  | 99.23   | 75.66  |
> | After Stage 2   | 4.57     | 4.13     | 4.21      | 62.84   | 69.65  | 85.49  | 69.60  | 63.17  | 99.42   | 78.71  |
> | Incremental Gains | +0.12 | +0.09     | +0.18     | +7.14   | +3.74  | −0.44  | +4.60  | +4.37  | +0.19   | +3.05  |
>
> *Observations and Analysis.*
> Stage 2 fine-tuning yields consistent gains across most subsets, with particularly notable improvements in *Web Questions*, *SD-QA*, *IFEval*, *BBH*, and *MMSU*. Based on manual inspection of the questions and model outputs in these subsets, we attribute the improvements to two main factors:
>
> 1. **Rare word / Further alignment.** Web Questions and SD-QA contain relatively infrequent terms. The Stage 1 model exhibits weaker alignment for such rare words, whereas Stage 2 fine-tuning improves the speech–text alignment for these cases.
> 2. **Enhanced utilization of LLM capabilities.** BBH, IFEval, and MMSU are strongly tied to the LLM's internal knowledge and reasoning abilities. Stage 2 end-to-end fine-tuning enables the speech tokenizer's audio embeddings to better match the LLM input space, allowing more effective invocation of these existing capabilities.
>
> We will move key parts of this analysis into the main text if space permits.

---

> ### Author Response · Authors · 2025-11-24
>
> **Part 3/3**
>
> ---
>
> **Question 2**: Tokenizer Difference
>
> > Qwen's tokenizer may segment differently from Whisper's. Did you observe systematic failure modes. Any language/domain-specific mitigations?
>
> **Response to Question 2**: We thank the reviewer for raising this insightful question. Indeed, the tokenizers of Qwen and Whisper can produce different segmentations for certain words, which in principle might lead to inconsistencies. However, in our experiments, we did *not observe any systematic failure modes* attributable to these differences. We believe that this is because the *two-stage fine-tuning procedure in AlignChat effectively transfers and adapts the pretrained capabilities* of the Whisper model to Qwen's tokenizer space.
>
> To further substantiate this, we conducted *AlignChat* using a different LLM backbone, *Llama‑3.1‑8B‑Instruct*, which has its own tokenizer. The OpenAudioBench results are shown below:
>
> | LLM Backbone            | AlpacaEval | Llama Questions | TriviaQA | Web Questions |
> | --------                | --------   | --------        | -------- | --------      |
> | Qwen2.5‑7B‑Instruct     | 81.26      | 80.33           | 61.40    | 65.90         |
> | Llama‑3.1‑8B‑Instruct   | 79.44      | 81.67           | 58.40    | 60.80         |
>
> > Due to the rebuttal time limit, we **do not** elaborate on the hyperparameters for Llama-3.1-8B-Instruct, which may cause the relatively inferior performance after switching the LLM backbone.
>
> We take these results as a hint that, despite LLM (*e.g.*, Qwen, Llama) and audio model (*e.g.*, Whisper) tokenizers potentially segmenting words differently, our two-stage fine-tuning effectively transfers capabilities across tokenizer spaces without systematic failure modes, and that the AlignChat training methodology generalizes well across **different LLM architectures and tokenizers**, preserving strong performance without requiring language- or domain-specific mitigation strategies.
>
> In the main text, we present results using Qwen2.5‑7B‑Instruct as our primary experimental model to ensure fair comparison with other baselines; nevertheless, when leveraging a more capable LLM, AlignChat may achieve even better performance on certain benchmarks.
>
> ---
>
> **Question 3**: Training Methods & Alignment Drift
>
> > How do you ensure $L _ {speech} \approx L _ {text}$ and maintain 1-to-1 correspondence without teacher forcing? Did you use scheduled sampling/professor forcing or any consistency regularizers? Please quantify alignment drift. How robust is the alignment under fillers, repetitions or partial words?
>
> **Response to Question 3**: We thank the reviewer for raising these important questions regarding maintaining 1‑to‑1 correspondence between speech and text tokens during inference. In our current training setup, we did **not** use scheduled sampling, professor forcing, or additional consistency regularizers. We attribute AlignChat's robust alignment primarily to the effective transfer of pretrained capabilities from the Whisper speech model and the target LLM within our two-stage fine-tuning framework.
>
> *Quantifying Alignment Drift.*
> To measure potential drift during inference, we randomly sampled 256 examples from the OpenAudioBench dataset and computed the **cosine similarity** and **L1 distance** between the audio embeddings (from the speech tokenizer) and the corresponding transcript embeddings (from the LLM tokenizer) at each token position. Full plots are provided in the revised appendix (Figure 8, Appendix).
>
> *Main observations:*
> - On average, **cosine similarity remained stable** with increasing token position, and **L1 distance did not show systematic growth** over the sequence.
> - For the longest sampled sequences, average cosine similarity declined from $\approx 0.8$ to $\approx 0.6$ towards the end; however, we suspect this is due to statistical noise from the small number of very long samples.
>
> Overall, these results suggest that **alignment drift is minimal** in our setting. We believe this robustness arises from the fact that speech and text embeddings are directly aligned: given an audio input, the speech tokenizer produces relatively fixed predictions, unlike in open-ended generation tasks (e.g., dialogue) where richer variability can lead to more pronounced drift.

---

### Author Response · Authors · 2025-12-01

Dear Reviewers, Area Chairs, and Program Chairs,

We sincerely thank all six reviewers for their constructive comments and insightful questions, which have helped us significantly refine our work.

---

Reviewers have acknowledged the impact, novelty, and firm performance of our proposed **AlignChat** method, as well as our comprehensive analysis.

**[Impactful and well‑motivated]**

- **Reviewer PPnd**: "Strict token‑level alignment between speech and text is a good step away from encoder‑only audio pipelines. The rate‑matching idea may generalize to other modalities."
- **Reviewer 8Srw**: "From the point of speech understanding, framing speech‑to‑text as a strict per‑token alignment problem is well motivated and interpretable."
- **Reviewer NsNf**: "The paper tackles an important problem, clearly formulated and mathematically sound."
- **Reviewer 8EeD**: "The authors build a good system with the proposed method, which shows good performance on 3 open‑source benchmarks."

**[Key Architectural Contributions]**

- **Reviewer XoS6**: "Introduces an encoder–decoder speech tokenizer that aligns speech tokens one‑to‑one with text embeddings; keeps the LLM frozen to avoid catastrophic forgetting."
- **Reviewer PPnd & 8Srw**: Recognized the value of sharing vocabulary and embedding layers between speech and text, and reducing audio token rate to lower computation without retraining the backbone.

**[Superior Performance & Data Efficiency]**

- **Reviewer zX26**: "AlignChat achieves improved performance on OpenAudioBench and VoiceBench using an order of magnitude less speech data (~15K hours), with ablations showing variance, generalization, and limitations."
- **Reviewer NsNf**: "Overall impressive results for speech‑to‑text chat tasks, with comprehensive ablations and representation visualizations."

---

During the response period, we carefully provided detailed feedback and conducted supplementary experiments to address all reviewer comments. We concisely summarize our responses here:

**[Presentation Improvements]**

- Clarified the role and training flow of the *modality projector*, ensuring a clear distinction between alignment loss application and ASR loss pathways.
- Revised imprecise statements to accurately reflect our design intent, *e.g., "projector alleviates conflict between transcription and alignment tasks."*
- Added explicit definitions for *encoder‑only vs. decoder‑only,* with references to representative systems (Kimi‑Audio, Qwen2.5‑Omni, LLaMA‑Omni).

**[Experiment Additions & Baselines]**

- Provided full details of the cascaded ASR + LLM baseline (*whisper‑large‑v3‑turbo* + *Qwen2.5‑7B-Instruct*), with comprehensive **VoiceBench** breakdown alongside commercial systems.
- Compared AlignChat with the cascaded ASR + LLM baseline across tasks, highlighting trade‑offs: cascaded yields higher scores on knowledge‑intensive/instruction tasks due to precise text representation, while AlignChat preserves non‑textual audio cues and achieves lower first‑token latency.
- Added **UnderEmotion‑en** benchmark results to demonstrate AlignChat's advantage in emotion‑sensitive dialogue (>6‑point gain over cascaded ASR + LLM).
- Extended OpenAudioBench analysis with error breakdown to distinguish speech tokenizer *vs.* LLM limitations.
- Conducted detailed ablation on L1 *vs.* cosine similarity alignment losses in Stage 1 & Stage 2, diagnosing performance collapse when removing either loss and discussing representational/gradient interference.
- Added embedding‑norm and cosine‑similarity histograms on OpenAudioBench; Mean cosine similarity $\approx 0.82$ shows strong directional alignment, while L1 loss constrains norms.
- Measured cosine and L1 alignment per token position on 256 random OpenAudioBench samples: minimal drift across sequences, with a slight drop on very long inputs likely due to sample sparsity.

**[Further Explanations]**

- Discussed the complementary roles of L1 and cosine similarity losses for stable alignment without LLM end-to-end supervision.
- Explained why AlignChat does not use native LLM text embeddings for speech input to avoid discarding non‑textual audio features, *a key differentiator from cascaded pipelines*.
- Described how AlignChat's strict rate‑matching and token alignment reduce computational overhead, enabling training with much less data while maintaining competitive accuracy.

---

We hope our detailed responses and new experiments address the reviewers' concerns. Your valuable feedback has helped us strengthen both the presentation and empirical validation of our work. We believe **AlignChat** provides a solid, extensible foundation for end–to–end speech-LLM integration.

Best regards and thanks,

Authors of #2926

---

### Meta-Review · Area_Chair_8N6n · 2025-12-14

**Summary:**

The paper proposes AlignChat, an end to end speech to text conversational framework that integrates speech understanding into a frozen LLM via strict token level alignment between speech and text. The key idea is to use a Whisper initialized encoder–decoder speech tokenizer whose output rate matches text tokens, enabling the model to share the LLM’s tokenizer, embeddings, and output head. Training proceeds in two stages: (1) ASR style alignment without running the LLM, and (2) instruction tuning using LLM generated targets while keeping the LLM frozen. Reported results show strong performance on OpenAudioBench and VoiceBench, with significantly less training data than prior work.

The reviewers highlight the data efficiency and competitive benchmark performance as notable strengths, along with ablations. However, they raise important concerns regarding novelty and the relationship to prior work, including BESTOW and earlier speech text alignment literature. More critically, the absence of comparisons with cascaded ASR–LLM pipelines and missing analyses of alignment quality and latency prevent reviewers from assessing where the claimed advantages come from. These concerns dominate the lower contributions.

**Reviewer Concerns:**

I don't think the reviewer concerns have been well addressed, such as the novelty and the comprehensive comparisons with cascaded ASR–LLM pipelines.

**Reviewer Scores:**

I don't think the reviewer would have changed their scores.

---

### Decision · Program_Chairs · 2026-01-26

Reject